# Towards Attributions of Input Variables in a Coalition

Xinhao Zheng [1]   Huiqi Deng [1]   Quanshi Zhang [1]

## Abstract

This paper focuses on the fundamental challenge of partitioning input variables in attribution methods for Explainable AI, particularly in Shapley value-based approaches. Previous methods always compute attributions given a predefined partition but lack theoretical guidance on how to form meaningful variable partitions. We identify that attribution conflicts arise when the attribution of a coalition differs from the sum of its individual variables' attributions. To address this, we analyze the numerical effects of AND-OR interactions in AI models and extend the Shapley value to a new attribution metric for variable coalitions. Our theoretical findings reveal that specific interactions cause attribution conflicts, and we propose three metrics to evaluate coalition faithfulness. Experiments on synthetic data, NLP, image classification, and the game of Go validate our approach, demonstrating consistency with human intuition and practical applicability.

## 1. Introduction

Estimating the attribution/importance/saliency of input variables (Selvaraju et al., 2017; Sundararajan et al., 2017; Lundberg & Lee, 2017) for an AI model represents one of the most typical direction in explainable AI. In particular, the Shapley value (Weber, 1988) is widely considered as a standard attribution method, because it is the unique attribution metric that satisfies the axioms of *anonymity, symmetry, dummy, additivity*, and *efficiency*.

However, many studies (Ren et al., 2023a; Li & Zhang, 2023) have pointed out that one major challenge in this area is how to define the partition of input variables. For example, there is no theory to determine whether to take pixels or local regions as input variables for image classification and take words or tokens as input variables for natural language

[1]Shanghai Jiao Tong University. Correspondence to: Quanshi Zhang <zqs1022@sjtu.edu.cn>.

*Proceedings of the $42^{nd}$ International Conference on Machine Learning*, Vancouver, Canada. PMLR 267, 2025. Copyright 2025 by the author(s).

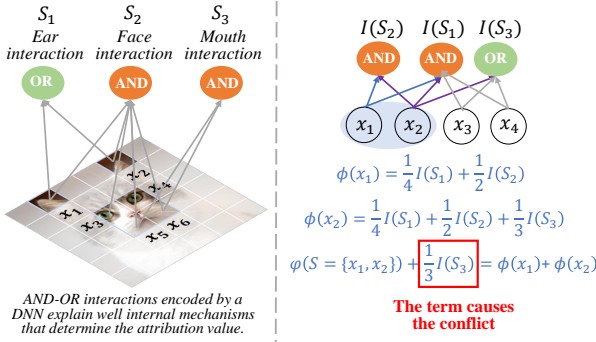

*Figure 1.* (a)AND-OR interaction: Let the AI model encode three interactions $S_1 = \{x_1, x_2\}$, $S_2 = \{x_1, x_2, x_3, x_4, x_5, x_6\}$ and $S_3 = \{x_5, x_6\}$, respectively. In this way, the Shapley value of $x_1$ can be decomposed as $\phi(x_1) = 1/2 \cdot I(S_1) + 1/6 \cdot I(S_2)$. (b) Conflict of attributions: Let us consider another example with three interactions, *w.r.t.*, $S_1 = \{x_1, x_2, x_3, x_4\}$, $S_2 = \{x_1, x_2\}$, and $S_3 = \{x_2, x_3, x_4\}$. The attribution of the coalition $\{x_1, x_2\}$ is not equal to the sum of attributions of input variable $x_1$ and $x_2$, *i.e.*, $\varphi(S = \{x_1, x_2\}) \neq \phi(x_1) + \phi(x_2)$.

processing. In other words, evaluating whether grouped variables in a partition can form a faithful basic unit or coalition, remains a challenge.

Essentially, this problem of the partition of input variables is rooted in the conflict of attributions computed under different partitions of input variables. Consider the full set of input variables $N = \{x_1, x_2, x_3, x_4\}$, as Figure 1(b) shows, we can directly compute the attribution of each $i$-th input variable, denoted by $\phi(i)$, under the partition $\{\{x_1\}, \{x_2\}, \{x_3\}, \{x_4\}\}$. Alternatively, we can also apply a new partition $\{\{x_1, x_2\}, \{x_3\}, \{x_4\}\}$ by grouping a set of input variables $\{x_1, x_2\}$, and we consider the entire set $\{x_1, x_2\}$ as a singleton variable, $S$, called *a coalition of input variables*. In this way, the attribution method may estimate the attribution of the coalition $S$, $\varphi(S)$. Thus, **the conflict of attributions means that the attribution of the coalition $S$ is not necessarily equal to the sum of attributions of the input variables in $S$, *i.e.*, $\varphi(S) \neq \sum_{i \in S} \phi(i)$.**

Therefore, the core task of this research is to derive the fundamental mechanism for such a conflict and to provide clearer guidance on whether variables grouped together can form a faithful basic unit or coalition.

**First, we discover the breakthrough point for proving the conflict is to disentangle all numerical effects in computing attributions.** We prove that the Shapley and Banzhaf values can be computed by reallocating the numerical effects of AND-OR interactions in the AI model to different input variables. The AND-OR interaction (Li & Zhang, 2023) measures the non-linear relationship between input variables, as shown in the toy example in Figure 1(a).

**Second, considering the interaction explains the internal mechanisms for the Shapley value, we extend the definition of the Shapley value and design a new attribution metric $\varphi(S)$ for a coalition $S$ of multiple variables. More importantly, we find that AND-OR interactions clarify the essential mechanism that causes the conflict between individual variables' attributions $\phi(i)$ and the coalition $S$'s attribution $\varphi(S)$.** Essentially, for each coalition $S$, the attribution of the coalition is computed using two types of interactions: (1) the interaction $T_1$ containing all variables in $S$ (*i.e.*, $T_1 \supseteq S$), and (2) the interaction $T_2$ containing partial variables in $S$ (*i.e.*, $T_2 \cap S \neq \emptyset$ and $T_2 \cap S \neq S$). The interaction $T_1$ is used to compute both Shapley values $\phi(i)$, $i \in S$, and the coalition attribution $\varphi(S)$. Whereas, the interaction $T_2$ is exclusively used to compute the Shapley value $\phi(i)$, $i \in S$. Specifically, the interaction $T_2$ is only used to compute the Shapley value $\phi(i)$, $i \in S$, but is not used to compute the attribution of the coalition $\varphi(S)$. Thus, the second type of interaction, *i.e.*, $T_2$ is the direct reason for the conflict of attributions.

**Third, we further propose three metrics to evaluate the faithfulness of the coalition and construct experiments under different scenarios.** Specifically, we evaluated our criteria on both synthetically generated function data and real-world tasks, including NLP and image classification, demonstrating consistency with human intuition. Furthermore, we applied our approach to the game of Go, where we found that it aligns with the understanding of Go players and can assist in discovering new interpretations of standard opening patterns.

**Contributions.** (1) We clarify the internal mechanism for the common conflict between individual variables' attributions and a coalition's attribution by using interactions to reformulate the attribution. (2) We propose a new coalition attribution metric with a clear explanation for such a conflict. (3) We further propose three metrics to assess coalition faithfulness and conduct experiments across various scenarios.

## 2. Related works

**Attribution methods.** Estimating the attribution of input variables for inference is a classic direction in the field of Explainable AI (Zhou et al., 2024; Ren et al., 2025). Some studies explained AI models using local decision bound-

aries (Ribeiro et al., 2016; Plumb et al., 2019), while others integrated gradients w.r.t. inputs (Sundararajan et al., 2017) or used gradient strength for feature attribution (Selvaraju et al., 2019). MUSE (Lakkaraju et al., 2019) analyzed decisions in lower-dimensional subspaces, and LRP (Bach et al., 2015) propagated model outputs to assign feature importance. DeepLIFT (Shrikumar et al., 2017) estimated input impacts by propagating differences from baseline predictions. Many works (Lundberg & Lee, 2017; Alshebli et al., 2019; Sundararajan & Najmi, 2020; Mitchell et al., 2022) used the Shapley value (Shapley et al., 1953), a standard attribution metric satisfying key axioms. (Covert et al., 2020) modified it for global efficiency, and (Mitchell et al., 2022) introduced efficient sampling methods. (Lundstrom & Razaviyayn, 2023) also proposed a unified framework for game-theoretic attribution methods, including the Shapley value, based on the Möbius transform. However, the above attribution methods do not focus on explaining the partition of input variables but rather compute the attribution value given a predefined partition.

**Interaction.** Unlike attribution methods estimating the importance of each input variable, the interaction usually provides a more fine-grained explanation, *i.e.*, the importance of each specific collaboration between a set of input variables. Some studies (Tsang et al., 2017) measured feature interactions in neural networks by considering salient weights between features. (Singh et al., 2019) created hierarchical explanations in feed-forward networks, while (Cui et al., 2019; Janizek et al., 2021) focused on pairwise interactions in Bayesian and second-order derivative neural networks. Inspired by the Shapley value, recent research measured feature group interactions by calculating marginal importance ($v(S) - v(\emptyset)$). (Sundararajan et al., 2020) introduced the Shapley Taylor interaction index, while (Lundberg et al., 2020) applied the Shapley interaction index (Grabisch & Roubens, 1999) to tree-based models. Archipelago (Tsang et al., 2020) provided a post-hoc explanation method, evaluating feature groups as a whole. (Tsai et al., 2023) proposed the Faithful Shapley Interaction index, extending the four standard Shapley axioms. (Harris et al., 2021) proposed the Joint Shapley value to assess feature sets' average contributions across different explanation orders. However, previous methods face the conflict between a coalition's attribution and the attributions of its individual variables (or sub-coalitions), in an engineering manner, without offering a theoretical explanation. Further explanation on coalitions and interactions can be found in Appendix A.

## 3. Algorithm

### 3.1. Preliminaries: AND-OR interactions

Given an AI model and an input sample $x = [x_1, x_2, ..., x_n]$ with $n$ input variables, $N = \{1, ..., n\}$ represents the index

set of all input variables and the model output on the input sample $x$ is represented as $v(x) \in \mathbb{R}$. There are various settings for $v(x)$ when the model has multiple output dimensions. In multi-category classification tasks, $v(x)$ is usually defined below, following (Deng et al., 2022).

$$v(x) = \log \frac{p(y = y^*|x)}{1 - p(y = y^*|x)} \quad (1)$$

where $y^*$ denotes the ground-truth label of the input $x$.

**Shapley value** (Shapley et al., 1953) is a well-known game-theoretic metric to measure the attribution/importance of each input variable to the output of the AI model. The Shapley value has been considered as the unique attribution method that satisfies the axioms of *anonymity, symmetry, dummy, additivity, and efficiency*. In this way, the Shapley value of the $i$-th variable is computed as follows:

$$\phi(i) = \sum_{S \subseteq N \setminus \{i\}} \frac{|S|!(n - |S| - 1)!}{n!} \cdot \left[ v(S \cup \{i\}) - v(S) \right] \quad (2)$$

where $| \cdot |$ denotes the cardinality of the set. Here, we use $v(S)$ to simplify the notation of $v(x_S)$, and $v(x_S)$ denotes the model output on a masked sample $x_S$. In the masked sample $x_S$, variables in $S$ are present, and variables in $N \setminus S$ are masked. In this way, $v(\emptyset)$ represents the model output when all input variables are masked, and $v(N)$ denotes the model output on the original input sample $x$.

**AND-OR interactions.** Given an AI model $v(\cdot)$ and a set $S \subseteq N$ ($S \neq \emptyset$), the AND interaction effect $I_{\text{and}}(S)$ and the OR interaction effect $I_{\text{or}}(S)$, between input variables in $S$ can be computed, as follows.

$$I_{\text{and}}(S) = \sum_{L \subseteq S} (-1)^{|S| - |L|} v_{\text{and}}(L) \quad (3)$$

$$I_{\text{or}}(S) = -\sum_{L \subseteq S} (-1)^{|S| - |L|} v_{\text{or}}(N \setminus L) \quad (4)$$

where $v_{\text{and}}(L) = 0.5v(L) + \gamma_L$ and $v_{\text{or}}(L) = 0.5v(L) - \gamma_L$ represent output components exclusively for AND interactions and OR interactions, respectively, subject to $v(L) = v_{\text{and}}(L) + v_{\text{or}}(L)$. In this way, the extraction of all AND-OR interactions is determined by learning optimal parameters $\{\gamma_L\}$. (Li & Zhang, 2023) have proposed to learn parameters $\{\gamma_L\}$ via a LASSO-like loss to achieve sparsest interactions, *i.e.*, $\sum_{S \subseteq N} |I_{\text{and}}(S)| + |I_{\text{or}}(S)|$.

**How to understand AND-OR interactions.** $I_{\text{and}}(S)$ measures the non-linear relationship (AND relationship) between input variables in $S \subseteq N$. The presence of all variables in $S$ contributes an effect $I_{\text{and}}(S)$ to the model's output. For example, we consider the slang term $S = \{x_1 = raining, x_2 = cats, x_3 = and, x_4 = dogs\}$ in the input sentence "*It was raining cats and dogs outside.*" An AI model may encode the AND relationship between the variables in $S$ as an inference pattern of "heavy rain." If all four words exist in the input sentence, the DNN will detect this inference pattern and make a numerical effect $I_{\text{and}}(S)$ to push the output towards the meaning of "heavy rain." The masking

of any word in $S$ will disrupt the AND relationship and remove the effect $I_{\text{and}}(S)$.

$I_{\text{or}}(S)$ represents the OR relationship among input variables in $S$. The presence of any variable in $S$ contributes an effect of $I_{\text{or}}(S)$ to the model's output. Given the sentence "*This movie is boring and disappointing*" for sentiment classification, the OR interaction between variables in $T = \{x_1 = boring, x_2 = disappointing\}$ represents a negative sentiment with $I_{\text{or}}(T) < 0$. If any words in $T$ are present, then this pattern is activated and contributes $I_{\text{or}}(T)$ to the model output $v(N)$. Only masking all words in $T$ can deactivate the interaction and remove the effect $I_{\text{or}}(T)$.

**Universal-matching property.** The output of an AI model can always be explained as numerical effects of AND-OR interactions. In particular, the faithfulness of interaction-based explanation is ensured by the universal-matching property (Li & Zhang, 2023). Given each randomly masked sample $x_S$ ($S \subseteq N$), we can always use AND-OR interactions to mimic the network output $v(S)$ on $x_S$. No matter how we randomly mask the input and obtain a masked sample $x_S$ *s.t.* $S \subseteq N$, the masking operation deactivates some AND-OR interactions, but it is proven that the network output on each randomly masked sample $v(S)$ can be always accurately estimated as the sum of numerical effects of the remaining AND-OR interactions. Please see Appendix B for details.

### 3.2. Revisiting attributions from interactions

In this section, we will revisit the conflict problem in the attribution method and reformulate classical attribution metrics from the perspective of interactions.

**What is a coalition?** Given an AI model $v$ with $n$ input variables in $N$, the estimation of the numerical attribution of each $i$-th input variable $\phi(i)$ depends on the partition of input variables. For example, given an input sentence "*raining cats and dogs*", some people choose to take each token as an input variable and compute the attributions of different tokens, $N = \{rain, -ing, cats, and, dogs\}$, while other people use words as input variables, $P = \{\{rain, -ing\}, cats, and, dogs\}$. **Thus, the combination of input variables can be considered as a coalition,** just like the coalition of $S = \{rain, -ing\}$ in the second case. As Figure 1 shows, people may manually select a set of input variables with actual semantics to construct a coalition.

Compared to estimating a scalar attribution of a coalition $S$, the interaction usually represents more disentangled effects. For example, for a coalition $S = \{a, b, c\}$, different interaction effects $I(\{a\}), I(\{b\}), I(\{c\}), I(\{a, b\}), I(\{a, c\}), I(\{b, c\}), I(\{a, b, c\})$ may affect the attribution of the coalition $S$. Please see Appendix A for a detailed comparison between coalition attribution and interaction effect.

**Conflict of attributions.** Definition 3.1 introduces the con-

flict of attributions between a coalition and variables.

**Definition 3.1.** Given two partitions of $n$ input variables $N = \{1, 2, ...., n\}$ and $P = \{S_1, S_2, ..., S_m\}$, subject to $N = \bigcup_{i=1}^{m} S_i, \ \forall i \neq j, \ S_i \cap S_j = \emptyset$, the conflict of attributions means that there exists a coalition $S_k$ such that the attribution of the coalition $S_k$ is not equal to the sum of attributions of its compositional variables, *i.e.* $\phi_P(S_k) \neq \sum_{i \in S_k} \phi_N(i)$.

The above conflict is common in practical applications. In particular, there does not exist a universally accepted partition of input variables. To this end, neither the Shapley value nor the Banzhaf value can eliminate the conflict of the attributions calculated under different partitions of variables.

**Reformulating attributions.** As the theoretical foundation to explain the conflict of attributions, let us first revisit classical game-theoretic attributions, such as the Shapley value (Shapley et al., 1953) and the Banzhaf value (Lehrer, 1988). We find that both two attribution metrics can be explained as an allocation of AND-OR interaction effects.

**Theorem 3.2.** *(Reformulation of the Shapley value, proved in Appendix C) The Shapley value $\phi(i)$ of each input variable $x_i$ can be explained as $\phi(i) = \sum_{S \subseteq N, i \in S} \frac{1}{|S|} [I_{and}(S) + I_{or}(S)]$.*

Theorem 3.2 explains the internal mechanism of the Shapley value, *i.e.*, the Shapley value is computed by evenly allocating each interaction effect to all its compositional input variables. For example, in the sentence "*It was raining cats and dogs outside.*", the DNN may encode the AND interaction $S_1 = \{x_1 = raining, x_2 = cats, x_3 = and, x_4 = dogs\}$ to represent a heavy rain. Because these four variables $x_1, x_2, x_3, x_4$ play the same role in the interaction $S_1$, it is supposed to allocate the numerical effect $I_{and}(S_1)$ uniformly to each variable, *i.e.*, allocating $\frac{1}{4} I_{and}(S_1)$ to the variable $x_2 = cats$. Besides, the variable $x_2 = cats$ may also be involved in other interactions, such as $I_{and}(S_2 = \{x_1, x_2\})$ and $I_{or}(S_3 = \{x_2, x_3, x_4\})$. Then the Shapley value of $x_2 = cats$ can be computed as the accumulation of all allocated effect $\phi(x_1) = \frac{1}{4} I_{and}(S_1) + \frac{1}{2} I_{and}(S_2) + \frac{1}{3} I_{or}(S_3)$.

**The Banzhaf value (Penrose, 1946)** is another classical attribution metric, and we find that the mechanism of the Banzhaf value can also be explained as a specific allocation of interaction effects. Specifically, the Banzhaf value of $i$-th variable is formulated as follows.

$$B(i) = \sum_{S \subseteq N \setminus \{i\}} \frac{1}{2^{|N|-1}} \cdot [v(S \cup \{i\}) - v(S)] \quad (5)$$

**Theorem 3.3.** *(Reformulation of the Banzhaf value, proved in Appendix D) The Banzhaf value $B(i)$ of each input variable $x_i$ can be reformulated as $B(i) = \sum_{S \subseteq N, i \in S} \frac{1}{2^{|S|-1}} [I_{and}(S) + I_{or}(S)]$.*

Theorem 3.3 shows that when we compute the Banzhaf

value of the $i$-th variable, the effects of the interactions $I_{and}(S)$ and $I_{or}(S)$ subject to $i \in S$ are allocated to $B(i)$ with a constant weight $\frac{1}{2^{|S|-1}}$.

### 3.3. Attribution value for a coalition

The Shapley value is widely recognized as a standard attribution method with a relatively solid theoretical foundation, and Section 3.2 has explained the internal mechanism for the Shapley value from the perspective of AND-OR interactions. Based on this, in this subsection, we further extend the Shapley value of each individual input variable to define the attribution of a coalition.

As mentioned in Definition 3.1, the core problem is that different partitions of input variables may lead to conflicts of attributions. To this end, previous studies usually apply an additional loss to force the method to extract attributions without suffering much from the conflict. For example, the Faith-Shap (Tsai et al., 2023) used the minimum squared loss to push attribution of a coalition $S$ towards the sum of attribution of individual variables in $S$, *i.e.*, pushing $v(S)$ towards $\sum_{i \in S} \phi(i)$. Please see Table 1 for a detailed comparison of previous methods.

As the same shown in Section 2, methods in Table 1 are designed to solve the conflict in an engineering manner. In comparison, we focus on the mathematical factor that causes the conflict of attributions. To this end, we are inspired by Theorem 3.2, which explains the Shapley value as a uniform re-allocation of each interaction effect $I_{and}(S)$ or $I_{or}(S)$ to each input variable $i$ in $S$. Just like that, we can similarly define the attribution of a coalition $S$, $\varphi(S)$, as the re-allocation of AND-OR interactions, as follows:

$$\forall S \subseteq N, \ \varphi(S) = \sum_{T \supseteq S} \frac{|S|}{|T|} [I_{and}(T) + I_{or}(T)] \quad (6)$$

For example, in an input sentence "*It was raining cats and dogs outside.*", we may annotate a coalition $S = \{x_1 = rain, x_2 = -ing\}$. Let us suppose that the model has encoded an AND interaction $T = \{x_1 = rain, x_2 = -ing, x_3 = cats, x_4 = and, x_5 = dogs\}$. The numerical effect $I_{and}(T)$ depends on all the five variables. Thus, $\frac{2}{5} I_{and}(T)$ is supposed to be assigned with the coalition S containing two input variables and be added to $\varphi(S)$. In this way, the attribution of the coalition $S$, $\varphi(S)$, can be explained as an allocation of all interactions $I_{and}(T)$ and $I_{or}(T)$ that cover all variables in $S$, *i.e.*, $T \supseteq S$.

### 3.4. Explaining the conflict of attributions

Instead of forcibly eliminating the conflict in an engineering manner (Tsai et al., 2023), we find that the conflict $\phi_P(S_k) \neq \sum_{i \in S_k} \phi_N(i)$ in Definition 3.1 naturally exists in different AI models. Thus, we need to face and accept such a conflict, instead of forcibly eliminating such an ob-

Table 1. Comparison between the solutions of the conflict of attributions in different attribution methods

| Attribution methods | Solutions for the conflict of attributions |
|---|---|
| Shapley value (Shapley et al., 1953) | Efficiency axiom $v(N) = \sum_{i \in N} \phi(i)$, but cannot ensure the efficiency property, *w.r.t.* any arbitrary set $S \subseteq N$, *i.e.*, $\varphi(S) \neq \sum_{i \in S} \phi(i)$ |
| Banzhaf value (Penrose, 1946) | 2-efficiency axiom: $B(i) + B(j) = B(\{i, j\})$ but do not satisfy $B(S) = \sum_{i \in S} B(i)$ |
| Joint Shapley value (Harris et al., 2021) | Joint linearity, dummy, efficiency, anonymity, symmetry axioms, but estimating the attribution of a set of features/interactions, like (Sundararajan et al., 2020) |
| Faith-Shap (Tsai et al., 2023) | Using a loss $\|v(S) - \sum_{i \in S} \phi(i)\|^2$ to alleviate the conflict |
| Our method | **Proving the conflict is naturally unavoidable, and quantifying the essential cause for the conflict** |

jective existence. **The theoretical explanation and clear disentanglement of effects responsible for the conflict is usually considered as a more faithful solution to the coalition's attribution.** To this end, Theorem 3.4 disentangles Shapley values of all input variables in S into two components, *i.e.*, (1) the attribution component $\phi_{\text{shared}}(S)$ shared by both the coalition and the individual variables, and (2) the conflicting attribution component $\phi_{\text{conflict}}(S)$.

**Theorem 3.4.** *(proved in Appendix E) For any coalition $S \subseteq N$, we have $\sum_{i \in S} \phi(i) = \phi_{shared}(S) + \phi_{conflict}(S)$. $\phi_{shared}(S) \stackrel{def}{=} \varphi(S)$ is the attribution component existing in both the coalition's attribution $\varphi(S)$ and individual input variable's attribution $\phi(i)$, thereby being termed the* shared attribution component. $\phi_{conflict}(S) = \sum_{T \subseteq N, T \cap S \neq \emptyset, T \cap S \neq S} \frac{|T \cap S|}{|T|} [I_{and}(T) + I_{or}(T)]$ *represents the conflict (or difference) between the coalition attribution and the individual variables' attribution.*

**Theorem 3.4 shows that the conflict between individual variables' attributions and the attribution of the coalition $S$ comes from numerical effects of all interactions $T$ that contain just partial but not all variables in $S$, subject to $\emptyset \neq T \cap S \neq S$.** This well fits the human understanding of the conflict, *i.e.*, not all interactions take S as a singleton coalition. In particular, Corollary 3.5 shows that if the DNN always considers all variables in $S$ as a coalition without encoding interactions covering partial but not all variables in $S$, *i.e.*, $\forall T \in \{T : S \not\subseteq T, T \subseteq N, i \in S\}$, $I_{\text{and}}(T) = I_{\text{or}}(T) = 0$, then there will be no conflict of attributions, *w.r.t.* the coalition $S$.

**Corollary 3.5.** *If a set of input variables in $S$ are always encoded by the DNN as a coalition without any interactions containing partial variables in $S$, i.e., $\forall T \in \{T : S \not\subseteq T, T \subseteq N, i \in S\}$, $I_{and}(T) = I_{or}(T) = 0$, then the attribution of the coalition $S$ can fully determine the Shapley value $\phi(i)$, i.e., $\phi(i) = \frac{1}{|S|}\varphi(S)$ and $\varphi(S) = \sum_{i \in S} \phi(i)$.*

**Explaining Shapley values.** Let us use the coalition attribution $\varphi(S)$ to explain the Shapley value of each variable $i$ in $S$, $\phi(i)$. Specifically, we can simply take the uniform al-

location of $\varphi(S)$ to its constituent input variables, $\frac{1}{|S|}\varphi(S)$, as the attribution of the $i$-th variable. Theorem 3.6 explains the difference between $\phi(i)$ and $\frac{1}{|S|}\varphi(S)$. It shows that the Shapley value $\phi(i)$ can be decomposed into two parts. (1) The first part $U_{i,S}$ is a component of coalition attribution $\frac{1}{|S|}\varphi(S)$, i.e., the uniform allocation of $\varphi(S)$ to its constituent input variables. (2) The second part $U_{i,\bar{S}}$ comes from all interactions $\{T\}$, which cover just partial but not all variables in $S$, *i.e.*, $\{T \mid T \subseteq N, i \in T \cap S, T \cap S \neq S\}$.

**Theorem 3.6.** *(proved in Appendix F) $\forall i \in S$,*
$\phi(i) = \sum_{T \subseteq N, T \supseteq S} \frac{1}{|T|} [I_{and}(T) + I_{or}(T)]$
$+ \sum_{T \subseteq N, T \not\supseteq S, T \ni i} \frac{1}{|T|} [I_{and}(T) + I_{or}(T)]$
$= \underbrace{\frac{1}{|S|}\varphi(S)}_{U_{i,S}} + \underbrace{\sum_{T \subseteq N, T \not\supseteq S, T \ni i} \frac{1}{|T|} [I_{and}(T) + I_{or}(T)]}_{U_{i,\bar{S}}}$

Corollary 3.7 further shows that if $S = \{i\}$ only contains a single variable $x_i$, then the attribution of the coalition $S$ is equal to the Shapley value $\phi(i)$.

**Corollary 3.7.** $\varphi(S = \{i\}) = \phi(i)$

**Verifying whether we can use coalition attributions to compute the Shapley value.** We conducted experiments to examine Theorem 3.6, *i.e.*, whether we could use the coalition $S$'s attribution to calculate the Shapley value of the $i$-th input variable, $i \in S$. We used $\Delta\phi(i)_S = \phi(i) - \hat{\phi}(i)$ to measure the approximation error between the true Shapley value computed based on Equation (2), $\phi(i)$, and the Shapley value estimated by Theorem 3.6, $\hat{\phi}(i)$. We conducted experiments on the DNNs introduced in Section 4.1. Table 2 reports the average estimation error $\mathbb{E}_{x,|S|=m}[\mathbb{E}|\Delta\phi(i)_S|]$ over all samples through all potential combinations of $(i, S)$ *s.t.*, $i \in S$ of a specific order $|S| = m$. The small errors proved the correctness of our theory.

### 3.5. Properties/axioms for the attribution of a coalition

Just like in the Shapley value, the following five axioms have been widely considered as standard requirements for reliable attributions (Lundberg & Lee, 2017). Although

there are slight differences in axioms for the Shapley value, the coalition attribution also satisfies these axioms, which are proven in Appendix G. Appendix H introduces a more detailed understanding of these axioms.

1. **Anonymity axiom:** Let us consider a permutation operation $\sigma$ on all input varaibles in $N$. $\sigma v$ denotes the permutated function subject to $\sigma v(\sigma(S)) = v(S)$, for all $S \subseteq N$ and $\sigma(S) = \{\sigma(i) : i \in S\}$. Then, it has $\varphi_v(S) = \varphi_{\sigma v}(\sigma(S))$.

2. **Symmetry axiom-$\alpha$:** If two input variables $i$ and $j$ have the same effect, *i.e.* $\forall S \subseteq N \backslash \{i, j\}, v(S \cup \{i\}) = v(S \cup \{j\})$, then $\forall S \subseteq N \backslash \{i, j\}, \varphi(S \cup \{i\}) = \varphi(S \cup \{j\})$.

3. **Symmetry axiom-$\beta$:** If two coalitions $S$ and $T$ ($|S| = |T|$) have the same effect, *i.e.* $\forall T' \subseteq T, \forall S' \subseteq S, |S'| = |T'|, \forall L \subseteq N \backslash (S' \cup T'), v(L \cup S') = v(L \cup T')$, then it has $\forall L \subseteq N \backslash (S \cup T), \varphi(L \cup S) = \varphi(L \cup T)$.

4. **Additivity axiom:** If the output of DNN can be divided into two independent parts, *i.e.* $\forall S \subseteq N, v(S) = v_1(S) + v_2(S)$, then the attribution of any coalition $S$ can also be decomposed to two parts, *i.e.* $\forall S \subseteq N, \varphi_v(S) = \varphi_{v_1}(S) + \varphi_{v_2}(S)$.

5. **Dummy axiom:** If a coalition $S$ is a dummy coalition, *i.e.* $\exists i \in S, \forall T \subseteq N \backslash \{i\}, v(T \cup \{i\}) = v(T)$, then it has no attribution to the DNN output, *i.e.* $\varphi(S) = 0$.

**Efficiency axiom. The overall model output can be decomposed into attributions of coalitions and attributions of individual variables.** Theorem 3.6 can directly derive the following corollary.

**Corollary 3.8.** *(Efficiency) For any coalition $S \subseteq N$, the output score of a model can be decomposed into the attribution of the coalition $S$ and the attribution of each input variable in $N \setminus S$ and the utilities of the interactions covering partial variables in $S$,* i.e.,
$$\forall S \subseteq N, v(N) - v(\emptyset) = \varphi(S) + \sum_{i \in N \setminus S} \phi(i) + \sum_{T \subseteq N, T \cap S \neq \emptyset, T \cap S \neq S} \frac{|T \cap S|}{|T|} [I_{and}(T) + I_{or}(T)]$$

Corollary 3.8 shows another type of efficiency property. I.e., an AI model's output on the input $x$ can be accurately mimicked by the sum of the coalition $S$'s attribution, the Shapley value of each variable in $N \setminus S$, and the numerical effects of the interaction covering partial variables in $S$.

**Verifying whether we can use coalition attributions to explain the network output.** According to Corollary 3.8, we can use the coalition attribution to mimic the output $v(N)$ of neural networks on any arbitrary sample. Therefore, given a coalition $S$, we followed Corollary 3.8 to compute $\hat{v}(N|S)$ to represent the model output mimicked by the coalition

*Table 2.* Approximate error $\mathbb{E}_{x, |S|=m}[\mathbb{E}|\Delta\phi(i)_S|]$ of using coalition attribution to mimic the Shapley value $\phi(i)$

| $m=1$ | $m=2$ | $m=3$ | $m=4$ | $m=5$ |
|---|---|---|---|---|
| $3.6 \times 10^{-8}$ | $1.1 \times 10^{-7}$ | $2.2 \times 10^{-7}$ | $4.2 \times 10^{-7}$ | $7.6 \times 10^{-7}$ |

| $m=6$ | $m=7$ | $m=8$ | $m=9$ | $m=10$ |
|---|---|---|---|---|
| $9.1 \times 10^{-7}$ | $4.4 \times 10^{-7}$ | $6.3 \times 10^{-7}$ | $8.8 \times 10^{-7}$ | $2.8 \times 10^{-7}$ |

*Table 3.* Approximate error $\mathbb{E}_{x, |S|=m}[\mathbb{E}|\Delta v_S|]$ of using coalition attribution to mimic the model output $v(N)$

| $m=1$ | $m=2$ | $m=3$ | $m=4$ | $m=5$ |
|---|---|---|---|---|
| $2.3 \times 10^{-7}$ | $5.8 \times 10^{-7}$ | $9.1 \times 10^{-7}$ | $1.8 \times 10^{-7}$ | $6.3 \times 10^{-7}$ |

| $m=6$ | $m=7$ | $m=8$ | $m=9$ | $m=10$ |
|---|---|---|---|---|
| $2.1 \times 10^{-7}$ | $5.1 \times 10^{-7}$ | $3.6 \times 10^{-7}$ | $3.1 \times 10^{-7}$ | $4.7 \times 10^{-7}$ |

attribution. We computed $\Delta v_S = v(N) - \hat{v}(N|S)$ to measure the approximation error. Table 3 reports the average estimation error $\mathbb{E}_{x, |S|=m}[\mathbb{E}|\Delta v_S|]$ over all samples. The settings of neural networks and datasets was introduced in Section 4.1. We found that no matter how the coalition $S$ was selected, the model output $v(N)$ could be well mimicked by the coalition attribution $\varphi(S)$ and the effect of interactions $T$ covering partial but not all variables in $S$.

## 4. Experiment

### 4.1. Evaluating faithfulness of a coalition

In this study, we prove the essential mechanism behind the conflicts of attributions computed on different partitions of input variables (see Theorem 3.6). **The decomposition of the Shapley value into two terms $U_{i,S}$ and $U_{i,\bar{S}}$ in Theorem 3.6 also enables us to evaluate the faithfulness of the coalition.** Specifically, (1) the $U_{i,S}$ term reflects the confidence of the coalition, because $U_{i,S}$ measures effects of all interactions $T$ ($T \supseteq S$) that take $S$ as a singleton variable; (2) In comparison, $U_{i,\bar{S}}$ reflects the significance of variables in $S$ that do not act as a singleton variable, because $U_{i,\bar{S}}$ measures the effect of interactions $T$ that cover just partial but not all variables in $S$. In this way, we propose three metrics to evaluate the faithfulness of the coalition.

The first metric $R(i)$ is designed to evaluate for each specific coalition $S \subseteq N$, whether the $U_{i,S}$ term dominates the major effect of $\phi(i)$. If so, we consider the set of variables in $S$ as a faithful coalition.

$$R(i) = \frac{|U_{i,S}|}{|U_{i,S}| + |U_{i,\bar{S}}|}, \quad i \in S \quad (7)$$

The second metric $R'(i) \in [0, 1]$ is defined to measure the significance of the variable $i$ participating in the coalition $S$, in a more fine-grained manner, as follows.

$$R'(i) = \frac{\sum_{T \supseteq S} \frac{1}{|T|}(|I_{and}(T)| + |I_{or}(T)|)}{\sum_{T' \ni i} \frac{1}{|T'|}(|I_{and}(T')| + |I_{or}(T')|)}, \quad i \in S \quad (8)$$

Table 4. Coalition faithfulness metrics on toy functions

|  | $\mathbb{E}_{f,i}[R(i)]$ | $\mathbb{E}_{f,i}[R'(i)]$ | $\mathbb{E}_f[Q(S)]$ |
|---|---|---|---|
| purely faithful coalitions | 0.944 | 0.936 | 0.948 |
| partially faithful coalitions | 0.471 | 0.608 | 0.590 |
| purely unfaithful coalitions | 0.031 | 0.016 | 0.013 |

where $\sum_{T \supseteq S} \frac{1}{|T|}(|I_{\text{and}}(T)| + |I_{\text{or}}(T)|)$ denotes the strength of interaction effects that are allocated from the coalition $S$ to the variable $i$, $i \in S$. $\sum_{T' \ni i} \frac{1}{|T'|}(|I_{\text{and}}(T')| + |I_{\text{or}}(T')|)$ denotes the strength of interaction effects allocated to the variable $i$, no matter whether or not the variable $i$ collaborates with other variables in $S \setminus \{i\}$ as a coalition.

Unlike the first two metrics focusing on the effect of a single variable $i$, the third metric $Q(S) \in [0, 1]$ is defined to measure the significance of the entire coalition $S$.

$$Q(S) = \frac{\sum_{T \supseteq S} \frac{|S|}{|T|}(|I_{\text{and}}(T)| + |I_{\text{or}}(T)|)}{\sum_{T' \subseteq N, T' \cap S \neq \emptyset} \frac{|T' \cap S|}{|T'|}(|I_{\text{and}}(T')| + |I_{\text{or}}(T')|)} \quad (9)$$

where the numerator measures the overall strength of effects allocated to the coalition $S$, and the denominator denotes the overall strength of effects of all variables in $S$, no matter whether these variables construct the entire coalition.

**Experiments on toy functions.** We conducted two experiments to use three metrics in Equations (7)-(9) to evaluate the faithfulness of different coalitions encoded by a DNN. However, the core problem was that it was difficult to obtain ground-truth interactions encoded by a DNN because true/ideal interactions that were supposed to be learned for a task were not necessarily equivalent to the real interactions that a DNN had learned. Thus, we trained DNNs to regress the following toy function $f(x) = \sum_{i=1}^{m} w_i \prod_{j \in T_i} x_j$ with clear interactions to determine ground-truth interactions, where $x = [x_1, x_2, ..., x_n] \in \{0, 1\}^n, \forall i \neq j, T_i \neq T_j$. The function $f(x)$ was determined by $m$ different true interactions $\{T_i | i = 1, 2, ..., m\}$. We designed a set of 20 target functions $f(x)$ for testing by applying different sets of $\{T_i\}$.

Specifically, we evaluated the faithfulness of the following three types of coalitions. **(1)** Given the target function $f(x)$, *w.r.t.* $m$ true interactions $\{T_i | i = 1, ..., m\}$, each of the first type of coalitions, $S$, was fully contained by some true interactions $T_i \supseteq S$, without being partially covered by any interactions $T_i$ ($T_i \cap S \neq \emptyset, T_i \cap S \neq S$). Thus, the first type of coalition was supposed to be the most **purely faithful coalitions**. **(2)** In contrast, each of the second type of coalition, $S$, was partially covered by some true interactions $T_i$ ($T_i \cap S \neq \emptyset, T_i \cap S \neq S$) without being fully contained by any interactions $T_i$. These coalitions were supposed to be **purely unfaithful coalitions**. **(3)** The third type of coalitions were termed **partially faithful coalitions**, *i.e.*, being contained by some interactions $T_i \supseteq S$ and partially covered by other interactions $T_j$. If the DNN was well trained to

fit the target function $f(x)$, then our metrics were supposed to identify these purely faithful/purely unfaithful/partially faithful coalitions.

In practice, we trained the MLP (Ren et al., 2023b) to regress the functions. Table 4 showed that the regression of a purely faithful coalition $S$ had $Q(S), R'(i), R(i)$ close to 1, and the regression of a purely unfaithful coalition $S$ always had $Q(S), R'(i), R(i)$ close to 0. A partially faithful coalition $S$ always had $0 < Q(S), R'(i), R(i) < 1$ which means that variable in $S$ existed in the true interactions containing the whole coalition $S$, and also existed in other interactions.

**Experiments on DNNs trained for NLP tasks.** We conducted experiments to evaluate whether the AI model faithfully encoded such natural coalitions in human cognition. We finetuned the pre-trained BERT-large (Devlin et al., 2018) and LLaMA (Touvron et al., 2023) model on SST-2 dataset (Socher et al., 2013) for sentiment classification. We divided the input sentence into words and manually selected 10 words as input variables. Given an input sentence $x$, we annotated some natural coalitions according to human cognition. For example, in the sentence "*It was raining cats and dogs outside,*" the phrase "*raining cats and dogs*" was annotated as a natural coalition. In comparison, we randomly selected coalitions as false coalitions.

Table 5 shows that in the sentence (a), the coalition $S$ "*mesmerizing performances*" represented the positive emotions with high $Q(S), R(i), R'(i)$ values on Bert-large and LLaMA models, thereby being considered as a faithful coalition. In comparison, in the sentence (b), the selected coalition $S$ "*rivaling blair*" separated the phrase "*blair witch,*" so $S$ was considered as an unfaithful coalition, which was also reflected by the low $Q(S), R(i), R'(i)$ values. It provides new insights that if a coalition contains both tokens related to the generated language and irrelevant tokens, then this coalition usually represents a representation flaw.

**Experiments on DNNs trained for image classification.** We conducted experiments on VGG-11 (Simonyan & Zisserman, 2014) and ResNet-20 (He et al., 2016) on the MNIST (LeCun et al., 1998) and CIFAR-10 (Krizhevsky, 2012) datasets and verified that these DNNs represented natural coalitions that fit human cognition. Please see Appendix I for experimental settings and results.

### 4.2. Application: explaining the Go game

Computing the attribution value of a coalition is of significant value in practice, and our method can be widely used. A typical application is to explain shape patterns memorized by a DNN to play the game of Go, which is inspired by the previous work (Zhou et al., 2023b). The shape patterns correspond to coalitions between input stones. Specifically, people usually use a value network to estimate the advantage

*Table 5.* Coalition attribution metrics on SST-2 dataset

| Sentences | Bert-large |
|---|---|
| (a) the **mesmerizing performances** of the leads keep the film grounded and keep the audience riveted. | $Q(\{\text{mesmerizing performances}\}) = 0.743$ 
 $R(\{\text{mesmerizing}\}) = 0.690, R'(\{\text{mesmerizing}\}) = 0.682$ 
 $R(\{\text{performances}\}) = 0.677, R'(\{\text{performances}\}) = 0.685$ |
| (b) one of creepiest, scariest movies to come along in a long, long time, easily **rivaling blair** witch or the others | $Q(\{\text{rivaling blair}\}) = 0.425$ 
 $R(\{\text{rivaling}\}) = 0.145, R'(\{\text{rivaling}\}) = 0.391$ 
 $R(\{\text{blair}\}) = 0.250, R'(\{\text{blair}\}) = 0.466$ |

| Sentences | LLaMA |
|---|---|
| (a) the **mesmerizing performances** of the leads keep the film grounded and keep the audience riveted. | $Q(\{\text{mesmerizing performances}\}) = 0.746$ 
 $R(\{\text{mesmerizing}\}) = 0.611, R'(\{\text{mesmerizing}\}) = 0.652$ 
 $R(\{\text{performances}\}) = 0.726, R'(\{\text{performances}\}) = 0.739$ |
| (b) one of creepiest, scariest movies to come along in a long, long time, easily **rivaling blair** witch or the others | $Q(\{\text{rivaling blair}\}) = 0.312$ 
 $R(\{\text{rivaling}\}) = 0.238, R'(\{\text{rivaling}\}) = 0.429$ 
 $R(\{\text{blair}\}) = 0.277, R'(\{\text{blair}\}) = 0.286$ |

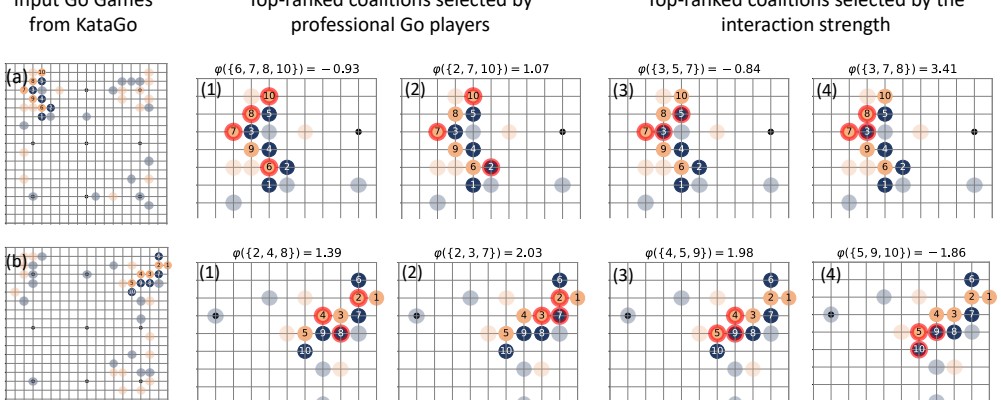

*Figure 2.* Visualization of two approaches for the selection of coalitions in KataGo. For a coalition $S$, $\varphi(S) > 0$ means the coalition $S$ of stones makes a positive numerical effect for the white, while it makes a negative effect when $\varphi(S) < 0$.

score of white stone in Go. The advantage score depends on complex coalitions (shape patterns) between white stones and black stones. However, there are no metrics to evaluate the true attribution of each shape pattern.

Therefore, we conducted experiments to quantify the attribution of each shape pattern encoded by the KataGo model (Wu, 2020), which was an open-source Go engine, and was known for its strong performance in playing the game of Go. The KataGo incorporated the Monte Carlo Tree Search and a value network to play the game, and we evaluated shape patterns encoded by the value network.

We had KataGo execute 40 moves against each other. Due to the significantly high computational cost of interactions, we only explained a local board state consisting of $n = 10$ stones (*i.e.*, 5 white stones and 5 black stones). These 10 stones were selected by professional Go players as input variables $N$, between which interactions were computed. The remaining stones on the board were treated as a constant background. Then, we followed (Zhou et al.,

2023a) to set $v(S) = \log(\frac{p_{\text{white}}(x_S)}{1-p_{\text{white}}(x_S)}) + a_k$, *s.t.* $a_k = \mathbb{E}_x \mathbb{E}_{T \subseteq N: k=n_{\text{white}}(T)-n_{\text{black}}(T)} \log(\frac{p_{\text{white}}(x_T)}{1-p_{\text{white}}(x_T)})$ measures the advantage score of white stones, where $n_{\text{white}}(T)$ is the number of white stones in $T$. $p_{\text{white}}(x_S)$ represents the score of white stones on input $x_S$, where stones in $N \setminus S$ were removed from the board. $a_k$, $k = -\frac{n}{2}, -\frac{n-2}{2}, ..., \frac{n}{2}$ represents a bias of the network output on a certain input.

Notably, although KataGo's capabilities surpassed those of human players, it was not necessary for the coalition/interactions modeled by KataGo to reflect human cognition of the Go game. Therefore, we used two strategies to select coalition candidates. The first strategy was to let professional human players annotate classical shape patterns, according to human understanding of the Go game. The second strategy was to select top-$k$ interactions selected by the interaction strength ($|I(T)|$) as coalition candidates suggested by the DNN. Then, given each coalition candidate, we calculated the attribution $\varphi(S)$ of selected coalitions. Figure 2 shows that coalitions that boosted the advantage score

of white stones *s.t.* $\varphi(S) > 0$ and coalitions that reduced the advantage score of white stones *s.t.* $\varphi(S) < 0$. These results help expert Go players learn shape patterns to play the Go game. Furthermore, we analyze the fitness between the extracted coalitions and human intuition on many more game boards. Please refer to Appendix J for details.

## 5. Conclusion

In this paper, we find and formulate the conflict between individual variables' attributions and the attribution of the coalition $S$. In order to explain the internal mechanism for such a conflict, we discover that both the Banzhaf value and the Shapley value can be formulated as a specific allocation of AND-OR interactions. Inspired by that, we extend AND-OR interactions to define the coalition attribution, and we prove that the conflict of attributions comes from numerical effects of all interactions $T$ that contain just partial but not all variables in $S$. Furthermore, the new coalition attribution enables us to evaluate the faithfulness of a coalition.

## Impact Statement

This paper aims to explain the internal mechanism for the conflict between individual variables' attributions and the attribution of the coalition $S$. We extend AND-OR interactions to define the coalition attribution, and we prove that the conflict of attributions comes from numerical effects of all interactions that contain just partial but not all variables in $S$. Furthermore, the new coalition attribution could help researchers evaluate the faithfulness of a coalition.

## Acknowledgements

This work is partially supported by the National Science and Technology Major Project (2021ZD0111602), the National Nature Science Foundation of China (92370115, 62276165), and Shanghai Natural Science Foundation (24ZR1491700).

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

## A. Coalition attribution vs interaction effect

Unlike the coalition attribution, interaction metrics (Harsanyi, 1959; Grabisch & Roubens, 1999; Sundararajan et al., 2020; Tsai et al., 2023) usually disentangle explicit effect $I(S)$ of each specific subset $S \subseteq N$ away from the effect $I(S')$ of its overlapping neighbor subset $S'$, instead of considering how to merge effects of overlapping subsets into the attribution of a coalition $T \subseteq S, S'$. For example, the Harsanyi interaction (Harsanyi, 1959) separately calculates the interaction effects $I(S_1 = \{x_1, x_2\})$, $I(S_2 = \{x_1, x_2, x_3\})$, and $I(S_3 = \{x_1, x_2, x_3, x_4\})$. There are no direct connections between these interaction effects. In contrast, the coalition attribution focuses on how to reasonably summarize these interaction effects into a single scalar importance score $\varphi(S_1)$ of the coalition $S_1$. In practice, both approaches can be applied to various types of applications, the analysis of byte blocks in cryptographic round functions (Zheng et al., 2024).

## B. Universal-matching property of AND-OR interactions

(Li & Zhang, 2023) have proven that the output of an AI model can always be explained by AND-OR interactions. For each input sample $x$, we can randomly mask $x$ and generate $2^n$ different masked samples $\{x_S\}$, *w.r.t.* $S \subseteq N$. Given each randomly masked sample $x_S$, we can always use AND-OR interactions to mimic the network output $v(S)$ on $x_S$, as follows.

$$v(S) = v(\emptyset) + \sum_{L \subseteq S, L \neq \emptyset} I_{\text{and}}(L) + \sum_{L \cap S \neq \emptyset, L \neq \emptyset} I_{\text{or}}(L) \tag{10}$$

Furthermore, (Ren et al., 2023a) have shown that the effects of interactions in most DNNs are usually very sparse. The majority of interaction effects are nearly zero, and only a few of the most salient interaction effects are sufficient to approximate the network's output $v(N)$.

## C. Proof of Theorem 2

*Proof.* According to the definition of Shapley values, we have: $\phi(i) = \sum_{S \subseteq N \setminus \{i\}} \frac{|S|!(n-|S|-1)!}{n!} \cdot \left[v(S \cup \{i\}) - v(S)\right] = \mathbb{E}_{S \subseteq N \setminus \{i\}}\left[v(S \cup \{i\}) - v(S)\right]$.

Then, according to Equation (10) in the paper, we have: $\forall S \subseteq N, v(S) = v(\emptyset) + \sum_{L \subseteq S, L \neq \emptyset} I_{\text{and}}(L) + \sum_{L \cap S \neq \emptyset} I_{\text{or}}(L)$. Thus, we have:

$$
\begin{aligned}
&v(S \cup \{i\}) - v(S) \\
&= \left[v(\emptyset) + \sum_{L \subseteq (S \cup \{i\}), L \neq \emptyset} I_{\text{and}}(L) + \sum_{L \cap (S \cup \{i\}) \neq \emptyset} I_{\text{or}}(L)\right] - \left[v(\emptyset) + \sum_{L \subseteq S, L \neq \emptyset} I_{\text{and}}(L) + \sum_{L \cap S \neq \emptyset} I_{\text{or}}(L)\right] \\
&= \left[\sum_{L \subseteq (S \cup \{i\}), L \neq \emptyset} I_{\text{and}}(L) - \sum_{L \subseteq S, L \neq \emptyset} I_{\text{and}}(L)\right] + \left[\sum_{L \cap (S \cup \{i\}) \neq \emptyset} I_{\text{or}}(L) - \sum_{L \cap S \neq \emptyset} I_{\text{or}}(L)\right] \\
&= \underbrace{\sum_{L \subseteq S} I_{\text{and}}(L \cup \{i\})}_{\mathcal{A}} + \underbrace{\sum_{L \cap S = \emptyset} I_{\text{or}}(L \cup \{i\})}_{\mathcal{B}}
\end{aligned}
$$

This allows us to break down the Shapley value into $\phi(i) = \mathbb{E}_{S \subseteq N \setminus \{i\}}[\mathcal{A} + \mathcal{B}]$.

In the subsequent proof, we first prove that the sum of AND interactions $\mathbb{E}_{S \subseteq N \setminus \{i\}}[\mathcal{A}]$ is equal to $\sum_{S \subseteq N, i \in S} \frac{1}{|S|} I_{\text{and}}(S)$.

$$\mathbb{E}_{S \subseteq N \setminus \{i\}}[\mathcal{A}]$$

$$= \mathbb{E}_{S \subseteq N \setminus \{i\}} \sum_{L \subseteq S} I_{\text{and}}(L \cup \{i\})$$

$$= \frac{1}{n} \sum_{m=0}^{n-1} \frac{1}{\binom{n-1}{m}} \sum_{\substack{S \subseteq N \setminus \{i\}, \\ |S|=m}} \sum_{L \subseteq S} I_{\text{and}}(L \cup \{i\})$$

$$= \frac{1}{n} \sum_{L \subseteq N \setminus \{i\}} \sum_{m=0}^{n-1} \frac{1}{\binom{n-1}{m}} \sum_{\substack{S \supseteq L, \\ S \subseteq N \setminus \{i\}, \\ |S|=m}} I_{\text{and}}(L \cup \{i\})$$

$$= \frac{1}{n} \sum_{L \subseteq N \setminus \{i\}} \sum_{m=|L|}^{n-1} \frac{1}{\binom{n-1}{m}} \sum_{\substack{S \supseteq L, \\ S \subseteq N \setminus \{i\}, \\ |S|=m}} I_{\text{and}}(L \cup \{i\}) \quad \text{// since } S \supseteq L, |S| = m \geq |L|.$$

$$= \frac{1}{n} \sum_{L \subseteq N \setminus \{i\}} \sum_{m=|L|}^{n-1} \frac{1}{\binom{n-1}{m}} \binom{n-1-|L|}{m-|L|} I_{\text{and}}(L \cup \{i\})$$

$$= \frac{1}{n} \sum_{L \subseteq N \setminus \{i\}} \underbrace{\sum_{k=0}^{n-1-|L|} \frac{1}{\binom{n-1}{|L|+k}} \binom{n-1-|L|}{k}}_{\alpha_L} I_{\text{and}}(L \cup \{i\}) \quad \text{// Let } k = m - |L|.$$

$$= \sum_{L \subseteq N \setminus \{i\}} \frac{1}{|L|+1} I_{\text{and}}(L \cup \{i\})$$

$$= \sum_{S \subseteq N, i \in S} \frac{1}{|S|} I_{\text{and}}(S) \quad \text{// Let } S = L \cup \{i\}.$$

Then, for the sum of OR interactions, we have

$$\mathbb{E}_{S \subseteq N \setminus \{i\}}[\mathcal{B}]$$

$$= \mathbb{E}_{S \subseteq N \setminus \{i\}} \sum_{L \cap S \neq \emptyset} I_{\mathrm{or}}(L \cup \{i\})$$

$$= \frac{1}{n} \sum_{m=0}^{n-1} \frac{1}{\binom{n-1}{m}} \sum_{\substack{S \subseteq N \setminus \{i\}, \ L \cap S \neq \emptyset \\ |S| = m}} \sum I_{\mathrm{or}}(L \cup \{i\})$$

$$= \frac{1}{n} \sum_{L \subseteq N \setminus \{i\}} \sum_{m=0}^{n-1} \frac{1}{\binom{n-1}{m}} \sum_{\substack{S \cap L \neq \emptyset, \\ S \subseteq N \setminus \{i\}, \\ |S| = m}} I_{\mathrm{or}}(L \cup \{i\})$$

$$= \frac{1}{n} \sum_{L \subseteq N \setminus \{i\}} \sum_{m=0}^{n-1} \frac{1}{\binom{n-1}{m}} \sum_{\substack{S \subseteq N \setminus \{i\} \setminus L, \\ |S| = m}} I_{\mathrm{or}}(L \cup \{i\})$$

$$= \frac{1}{n} \sum_{L \subseteq N \setminus \{i\}} \sum_{m=0}^{n-1-|L|} \frac{1}{\binom{n-1}{m}} \sum_{\substack{S \subseteq N \setminus \{i\} \setminus L, \\ |S| = m}} I_{\mathrm{or}}(L \cup \{i\}) \quad \text{// Since } S \subseteq N \setminus \{i\} \setminus L, |S| \leq n - 1 - |L|.$$

$$= \frac{1}{n} \sum_{L \subseteq N \setminus \{i\}} \sum_{m=0}^{n-1-|L|} \frac{1}{\binom{n-1}{m}} \binom{n-1-|L|}{m} I_{\mathrm{or}}(L \cup \{i\})$$

$$= \frac{1}{n} \sum_{L \subseteq N \setminus \{i\}} \sum_{k=0}^{n-1-|L|} \frac{1}{\binom{n-1}{n-1-|L|-k}} \binom{n-1-|L|}{n-1-|L|-k} I_{\mathrm{or}}(L \cup \{i\}) \quad \text{// Let } k = n - 1 - |L| - m.$$

$$= \frac{1}{n} \sum_{L \subseteq N \setminus \{i\}} \sum_{k=0}^{n-1-|L|} \frac{1}{\binom{n-1}{|L|+k}} \binom{n-1-|L|}{k} I_{\mathrm{or}}(L \cup \{i\})$$

$$= \frac{1}{n} \sum_{L \subseteq N \setminus \{i\}} \frac{n}{|L| + 1} I_{\mathrm{or}}(L \cup \{i\})$$

$$= \sum_{L \subseteq N \setminus \{i\}} \frac{1}{|L| + 1} I_{\mathrm{or}}(L \cup \{i\})$$

$$= \sum_{S \subseteq N, i \in S} \frac{1}{|S|} I_{\mathrm{or}}(S) \quad \text{// Let } S = L \cup \{i\}.$$

Therefore, $\phi(i) = \sum_{S \subseteq N \setminus \{i\}}[\mathcal{A}] + \sum_{S \subseteq N \setminus \{i\}}[\mathcal{B}] = \sum_{S \subseteq N, i \in S} \frac{1}{|S|}[I_{\mathrm{and}}(S) + I_{\mathrm{or}}(S)]$. $\qquad \square$

## D. Proof of Theorem 3

*Proof.* According to the definition of the AND/OR interaction, we can get:

$$I_{\mathrm{and}}(S) = \sum_{L \subseteq S}(-1)^{|S|-|L|} v_{\mathrm{and}}(L) = \sum_{L \subseteq S \setminus \{i\}}(-1)^{|S|-|L|+1} [v_{\mathrm{and}}(L \cup \{i\}) - v_{\mathrm{and}}(L)]$$

$$I_{\mathrm{or}}(S) = -\sum_{L \subseteq S}(-1)^{|S|-|L|} v_{\mathrm{or}}(N \setminus L) = \sum_{L \subseteq S \setminus \{i\}}(-1)^{|S|-|L|+1} [v_{\mathrm{or}}(N \setminus L) - v_{\mathrm{or}}(N - L - \{i\})]$$

where $v(L) = v_{\mathrm{and}}(L) + v_{\mathrm{or}}(L)$.

Then we have:

$$\sum_{S \subseteq N, S \ni i} \frac{1}{2^{|S|-1}} \left[ I_{\text{and}}(S) + I_{\text{or}}(S) \right]$$

$$= \sum_{S \subseteq N, S \ni i} \sum_{L \subseteq S \setminus \{i\}} \frac{(-1)^{|S|-|L|+1}}{2^{|S|-1}} \left\{ [v_{\text{and}}(L \cup \{i\}) - v_{\text{and}}(L)] + [v_{\text{or}}(N \setminus L) - v_{\text{or}}(N - L - \{i\})] \right\}$$

$$= \sum_{S \subseteq N \setminus \{i\}} \sum_{L \subseteq S} \frac{(-1)^{|S|-|L|}}{2^{|S|}} \left\{ [v_{\text{and}}(L \cup \{i\}) - v_{\text{and}}(L)] + [v_{\text{or}}(N \setminus L) - v_{\text{or}}(N - L - \{i\})] \right\}$$

$$= \sum_{L \subseteq N \setminus \{i\}} (-1)^{|L|} \sum_{S \subseteq N \setminus \{i\}, S \supseteq L} \frac{(-1)^{|S|}}{2^{|S|}} \left\{ [v_{\text{and}}(L \cup \{i\}) - v_{\text{and}}(L)] + [v_{\text{or}}(N \setminus L) - v_{\text{or}}(N - L - \{i\})] \right\}$$

$$= \sum_{L \subseteq N \setminus \{i\}} (-1)^{|L|} \frac{(-1)^{|L|}}{2^{|N|-1}} \left\{ [v_{\text{and}}(L \cup \{i\}) - v_{\text{and}}(L)] + [v_{\text{or}}(N \setminus L) - v_{\text{or}}(N - L - \{i\})] \right\}$$

$$\mathbin{/\!/} \sum_{S \subseteq N \setminus \{i\}, S \supseteq L} \frac{(-1)^{|S|}}{2^{|S|}} = \frac{(-1)^{|L|}}{2^{|N|-1}}.$$

$$= \sum_{L \subseteq N \setminus \{i\}} \frac{1}{2^{|N|-1}} \left\{ [v_{\text{and}}(L \cup \{i\}) - v_{\text{and}}(L)] + [v_{\text{or}}(N \setminus L) - v_{\text{or}}(N - L - \{i\})] \right\}$$

$$= \sum_{S \subseteq N \setminus \{i\}} \frac{1}{2^{|N|-1}} [v_{\text{and}}(S \cup \{i\}) - v_{\text{and}}(S)] + \sum_{S \subseteq N \setminus \{i\}} \frac{1}{2^{|N|-1}} [v_{\text{or}}(N \setminus S) - v_{\text{or}}(N - S - \{i\})] \quad \mathbin{/\!/} \text{ Let } L = S.$$

$$= \sum_{S \subseteq N \setminus \{i\}} \frac{1}{2^{|N|-1}} [v_{\text{and}}(S \cup \{i\}) - v_{\text{and}}(S)] + \sum_{S \subseteq N \setminus \{i\}} \frac{1}{2^{|N|-1}} [v_{\text{or}}(S \cup \{i\}) - v_{\text{or}}(S)]$$

$$= \sum_{S \subseteq N \setminus \{i\}} \frac{1}{2^{|N|-1}} \left\{ [v_{\text{and}}(S \cup \{i\}) + v_{\text{or}}(S \cup \{i\})] - [v_{\text{and}}(S) + v_{\text{or}}(S)] \right\}$$

$$= \sum_{S \subseteq N \setminus \{i\}} \frac{1}{2^{|N|-1}} \cdot [v(S \cup \{i\}) - v(S)]$$

$$= B(i)$$

Therefore, $B(i) = \sum_{S \subseteq N, S \ni i} \frac{1}{2^{|S|-1}} \left[ I_{\text{and}}(S) + I_{\text{or}}(S) \right]$. $\qquad \square$

## E. Proof of Theorem 4 & Corollary 5

*Proof.* According to Theorem 2, we have: $\phi(i) = \sum_{T \subseteq N, i \in T} \frac{1}{|T|} [I_{\text{and}}(T) + I_{\text{or}}(T)]$.

Then, according to Equation (6) in the paper, we have: $\varphi(S) = \sum_{T \supseteq S} \frac{|S|}{|T|} [I_{\text{and}}(T) + I_{\text{or}}(T)]$.

Thus, we have:

$$\sum_{i \in S} \phi(i)$$

$$= \sum_{i \in S} \sum_{T \subseteq N, T \ni i} \frac{1}{|T|} [I_{\text{and}}(T) + I_{\text{or}}(T)]$$

$$= \sum_{i \in S} \sum_{T \subseteq N, T \supseteq S} \frac{1}{|T|} [I_{\text{and}}(T) + I_{\text{or}}(T)] + \sum_{T \subseteq N, T \not\supseteq S, T \ni i} \frac{1}{|T|} [I_{\text{and}}(T) + I_{\text{or}}(T)]$$

$$= \sum_{i \in S} \left( \frac{1}{|S|} \sum_{T \subseteq N, T \supseteq S} \frac{|S|}{|T|} [I_{\text{and}}(T) + I_{\text{or}}(T)] + \sum_{T \subseteq N, T \not\supseteq S, T \ni i} \frac{1}{|T|} [I_{\text{and}}(T) + I_{\text{or}}(T)] \right)$$

$$= \sum_{i \in S} \left( \frac{1}{|S|} \varphi(S) + \sum_{T \subseteq N, T \not\supseteq S, T \ni i} \frac{1}{|T|} [I_{\mathrm{and}}(T) + I_{\mathrm{or}}(T)] \right)$$

$$= \sum_{i \in S} \frac{1}{|S|} \varphi(S) + \sum_{i \in S} \sum_{\substack{T \subseteq N, \\ T \not\supseteq S, \\ T \ni i}} \frac{1}{|T|} [I_{\mathrm{and}}(T) + I_{\mathrm{or}}(T)]$$

$$= |S| \cdot \frac{1}{|S|} \varphi(S) + \sum_{T \subseteq N, T \cap S \neq \emptyset, T \cap S \neq S} \frac{|T \cap S|}{|T|} [I_{\mathrm{and}}(T) + I_{\mathrm{or}}(T)]$$

// For any $T$, $\frac{1}{|T|}[I_{\mathrm{and}}(T) + I_{\mathrm{or}}(T)]$ will only be counted $|T \cap S|$ times.

$$= \varphi(S) + \sum_{T \subseteq N, T \cap S \neq \emptyset, T \cap S \neq S} \frac{|T \cap S|}{|T|} [I_{\mathrm{and}}(T) + I_{\mathrm{or}}(T)]$$

Therefore, we prove that: $\sum_{i \in S} \phi(i) = \varphi(S) + \sum_{T \subseteq N, T \cap S \neq \emptyset, T \cap S \neq S} \frac{|T \cap S|}{|T|} [I_{\mathrm{and}}(T) + I_{\mathrm{or}}(T)]$

Especially, if a set of input variables in $S$ are always encoded by the DNN as a coalition without any interactions containing partial variables in $S$, *i.e.*, $\forall T \in \{T : S \not\subseteq T, T \subseteq N, i \in S\}$, $I_{\mathrm{and}}(T) = I_{\mathrm{or}}(T) = 0$, then we have:

$$\sum_{i \in S} \phi(i) = \varphi(S) + \sum_{T \subseteq N, T \cap S \neq \emptyset, T \cap S \neq S} \frac{|T \cap S|}{|T|} [I_{\mathrm{and}}(T) + I_{\mathrm{or}}(T)] = \varphi(S)$$

Besides, we have:

$$\phi(i) = \frac{1}{|S|} \varphi(S) + \sum_{T \subseteq N, T \not\supseteq S, T \ni i} \frac{1}{|T|} [I_{\mathrm{and}}(T) + I_{\mathrm{or}}(T)] = \frac{1}{|S|} \varphi(S)$$

Therefore, we further prove Corollary 3.5. $\square$

## F. Proof of Theorem 6 & Corollary 7

*Proof.* According to Theorem 3.2, we have: $\phi(i) = \sum_{T \subseteq N, i \in T} \frac{1}{|T|} [I_{\mathrm{and}}(T) + I_{\mathrm{or}}(T)]$.

Then, according to Equation (6) in the paper, we have: $\varphi(S) = \sum_{T \supseteq S} \frac{|S|}{|T|} [I_{\mathrm{and}}(T) + I_{\mathrm{or}}(T)]$.

Thus, we have: $\forall i \in S$,

$$\phi(i)$$
$$= \sum_{T \subseteq N, T \ni i} \frac{1}{|T|} [I_{\mathrm{and}}(T) + I_{\mathrm{or}}(T)]$$
$$= \sum_{T \subseteq N, T \supseteq S} \frac{1}{|T|} [I_{\mathrm{and}}(T) + I_{\mathrm{or}}(T)] + \sum_{T \subseteq N, T \not\supseteq S, T \ni i} \frac{1}{|T|} [I_{\mathrm{and}}(T) + I_{\mathrm{or}}(T)]$$
$$= \frac{1}{|S|} \sum_{T \subseteq N, T \supseteq S} \frac{|S|}{|T|} [I_{\mathrm{and}}(T) + I_{\mathrm{or}}(T)] + \sum_{T \subseteq N, T \not\supseteq S, T \ni i} \frac{1}{|T|} [I_{\mathrm{and}}(T) + I_{\mathrm{or}}(T)]$$
$$= \frac{1}{|S|} \varphi(S) + \sum_{T \subseteq N, T \not\supseteq S, T \ni i} \frac{1}{|T|} [I_{\mathrm{and}}(T) + I_{\mathrm{or}}(T)]$$

Therefore, we prove that $\phi(i) = \frac{1}{|S|} \varphi(S) + \sum_{T \subseteq N, T \not\supseteq S, T \ni i} \frac{1}{|T|} [I_{\mathrm{and}}(T) + I_{\mathrm{or}}(T)]$.

Specially, we let $S = \{i\}$ and then we have:

$$\phi(i)$$
$$= \frac{1}{|S|}\varphi(S) + \sum_{T \subseteq N, T \not\supseteq S, T \ni i} \frac{1}{|T|}[I_{\text{and}}(T) + I_{\text{or}}(T)]$$
$$= \frac{1}{1}\varphi(S) + \sum_{T \subseteq N, T \not\supseteq \{i\}, T \ni i} \frac{1}{|T|}[I_{\text{and}}(T) + I_{\text{or}}(T)]$$
$$= \varphi(S) = \varphi(S = \{i\})$$

Therefore, we further prove Corollary 3.7: $\varphi(S = \{i\}) = \phi(i)$. □

## G. Proofs of axioms for the attribution of a coalition

In this section, we will prove the axioms for the attribution of a coalition in the main paper and Corollary 8.

### G.1. Proof of Anonymity axiom

*Proof.* According to the definition of the AND/OR interaction, we can get:

$I_{\text{and}_v}(T) = \sum_{L \subseteq T}(-1)^{|T|-|L|}v_{\text{and}}(L)$, $I_{\text{or}_v}(T) = -\sum_{L \subseteq T}(-1)^{|T|-|L|}v_{\text{or}}(N \setminus L)$

$I_{\text{and}_{\sigma v}}(\sigma(T)) = \sum_{L \subseteq \sigma(T)}(-1)^{|\sigma(T)|-|L|}\sigma v_{\text{and}}(L)$,

$I_{\text{or}_{\sigma v}}(\sigma(T)) = -\sum_{L \subseteq \sigma(T)}(-1)^{|\sigma(T)|-|L|}\sigma v_{\text{or}}(N \setminus L)$

Due to $\sigma v(\sigma(S)) = v(S)$, we have: $\sigma v_{\text{and}}(\sigma(S)) = v_{\text{and}}(S)$ and $\sigma v_{\text{or}}(\sigma(S)) = v_{\text{or}}(S)$.

Thus, we have:

$$I_{\text{and}_{\sigma v}}(\sigma(T)) = \sum_{L \subseteq \sigma(T)}(-1)^{|\sigma(T)|-|L|}\sigma v_{\text{and}}(L)$$
$$= \sum_{L = \sigma(K) \subseteq \sigma(T)}(-1)^{|\sigma(T)|-|\sigma(K)|}\sigma v_{\text{and}}(\sigma(K))$$
$$= \sum_{L = \sigma(K) \subseteq \sigma(T)}(-1)^{|T|-|K|}v_{\text{and}}(K)$$
$$= \sum_{K \subseteq T}(-1)^{|T|-|K|}v_{\text{and}}(K) = I_{\text{and}_v}(T)$$

$$I_{\text{or}_{\sigma v}}(\sigma(T)) = -\sum_{L \subseteq \sigma(T)}(-1)^{|\sigma(T)|-|L|}\sigma v_{\text{or}}(N \setminus L)$$
$$= -\sum_{L = \sigma(K) \subseteq \sigma(T)}(-1)^{|\sigma(T)|-|\sigma(K)|}\sigma v_{\text{or}}(N \setminus \sigma(K))$$
$$= -\sum_{L = \sigma(K) \subseteq \sigma(T)}(-1)^{|\sigma(T)|-|\sigma(K)|}\sigma v_{\text{or}}(\sigma(N \setminus K))$$
$$= -\sum_{K \subseteq T}(-1)^{|T|-|K|}v_{\text{or}}(N \setminus K) = I_{\text{or}_v}(T)$$

Then, according to Equation (6) in the paper, we have: $\varphi_v(S) = \sum_{T \supseteq S} \frac{|S|}{|T|}[I_{\text{and}_v}(T) + I_{\text{or}_v}(T)]$ and $\varphi_{\sigma v}(\sigma(S)) = \sum_{T \supseteq \sigma(S)} \frac{|\sigma(S)|}{|T|}[I_{\text{and}_{\sigma v}}(T) + I_{\text{or}_{\sigma v}}(T)]$.

Thus, we have:

$$\varphi_{\sigma v}(\sigma(S)) = \sum_{T \supseteq \sigma(S)} \frac{|\sigma(S)|}{|T|}[I_{\text{and}_{\sigma v}}(T) + I_{\text{or}_{\sigma v}}(T)] = \sum_{T = \sigma(L) \supseteq \sigma(S)} \frac{|\sigma(S)|}{|\sigma(L)|}[I_{\text{and}_{\sigma v}}(\sigma(L)) + I_{\text{or}_{\sigma v}}(\sigma(L))]$$
$$= \sum_{T = \sigma(L) \supseteq \sigma(S)} \frac{|S|}{|L|}[I_{\text{and}_v}(L) + I_{\text{or}_v}(L)] = \sum_{L \supseteq S} \frac{|S|}{|L|}[I_{\text{and}_v}(L) + I_{\text{or}_v}(L)] = \varphi_v(S)$$

Therefore, we prove the Anonymity axiom. □

### G.2. Proof of Symmetry axiom-$\alpha$

*Proof.* According to the definition of the AND/OR interaction, we can get:

$$I_{\text{and}}(T \cup \{i\}) = \sum_{L \subseteq (T \cup \{i\})} (-1)^{|T \cup \{i\}| - |L|} v_{\text{and}}(L), \quad I_{\text{or}}(T \cup \{i\}) = -\sum_{L \subseteq (T \cup \{i\})} (-1)^{|T \cup \{i\}| - |L|} v_{\text{or}}(N \setminus L)$$

$$I_{\text{and}}(T \cup \{j\}) = \sum_{L \subseteq (T \cup \{j\})} (-1)^{|T \cup \{j\}| - |L|} v_{\text{and}}(L), \quad I_{\text{or}}(T \cup \{j\}) = -\sum_{L \subseteq (T \cup \{j\})} (-1)^{|T \cup \{j\}| - |L|} v_{\text{or}}(N \setminus L)$$

Due to $\forall S \subseteq N \setminus \{i, j\}, v(S \cup \{i\}) = v(S \cup \{j\})$, we have: $\forall S \subseteq N \setminus \{i, j\}, v_{\text{and}}(S \cup \{i\}) = v_{\text{and}}(S \cup \{j\}), v_{\text{or}}(S \cup \{i\}) = v_{\text{or}}(S \cup \{j\})$

Then, we have:

$$
\begin{aligned}
& I_{\text{and}}(T \cup \{i\}) - I_{\text{and}}(T \cup \{j\}) \\
=& \sum_{L \subseteq (T \cup \{i\})} (-1)^{|T \cup \{i\}| - |L|} v_{\text{and}}(L) - \sum_{L \subseteq (T \cup \{j\})} (-1)^{|T \cup \{j\}| - |L|} v_{\text{and}}(L) \\
=& \left( \sum_{L \subseteq T} (-1)^{|T \cup \{i\}| - |L|} v_{\text{and}}(L) + \sum_{L \subseteq T} (-1)^{|T \cup \{i\}| - |L \cup \{i\}|} v_{\text{and}}(L \cup \{i\}) \right) \\
& - \left( \sum_{L \subseteq T} (-1)^{|T \cup \{j\}| - |L|} v_{\text{and}}(L) + \sum_{L \subseteq T} (-1)^{|T \cup \{j\}| - |L \cup \{j\}|} v_{\text{and}}(L \cup \{j\}) \right) \\
=& \sum_{L \subseteq T} (-1)^{|T| - |L|} [v_{\text{and}}(L \cup \{i\}) - v_{\text{and}}(L \cup \{j\})] = 0
\end{aligned}
$$

$$
\begin{aligned}
& I_{\text{or}}(T \cup \{i\}) - I_{\text{or}}(T \cup \{j\}) \\
=& -\sum_{L \subseteq (T \cup \{i\})} (-1)^{|T \cup \{i\}| - |L|} v_{\text{or}}(N \setminus L) + \sum_{L \subseteq (T \cup \{j\})} (-1)^{|T \cup \{j\}| - |L|} v_{\text{or}}(N \setminus L) \\
=& \left( \sum_{L \subseteq T} (-1)^{|T \cup \{j\}| - |L|} v_{\text{or}}(N \setminus L) + \sum_{L \subseteq T} (-1)^{|T \cup \{j\}| - |L \cup \{j\}|} v_{\text{or}}(N \setminus (L \cup \{j\})) \right) \\
& - \left( \sum_{L \subseteq T} (-1)^{|T \cup \{i\}| - |L|} v_{\text{or}}(N \setminus L) + \sum_{L \subseteq T} (-1)^{|T \cup \{i\}| - |L \cup \{i\}|} v_{\text{or}}(N \setminus (L \cup \{i\})) \right) \\
=& \sum_{L \subseteq T} (-1)^{|T| - |L|} [v_{\text{or}}((N \setminus (L \cup \{i, j\})) \cup \{i\}) - v_{\text{or}}((N \setminus (L \cup \{i, j\})) \cup \{j\})] = 0
\end{aligned}
$$

Besides, according to Equation (6) in the paper, we have: $\varphi(S \cup \{i\}) = \sum_{T \supseteq (S \cup \{i\})} \frac{|S \cup \{i\}|}{|T|} [I_{\text{and}}(T) + I_{\text{or}}(T)]$ and $\varphi(S \cup \{j\}) = \sum_{T \supseteq (S \cup \{j\})} \frac{|S \cup \{j\}|}{|T|} [I_{\text{and}}(T) + I_{\text{or}}(T)]$.

Then, we have:

$$
\begin{aligned}
& \varphi(S \cup \{i\}) - \varphi(S \cup \{j\}) \\
=& \sum_{T \supseteq (S \cup \{i\})} \frac{|S \cup \{i\}|}{|T|} [I_{\text{and}}(T) + I_{\text{or}}(T)] - \sum_{T \supseteq (S \cup \{j\})} \frac{|S \cup \{j\}|}{|T|} [I_{\text{and}}(T) + I_{\text{or}}(T)] \\
=& \left( \sum_{T \supseteq (S \cup \{i,j\})} \frac{|S| + 1}{|T|} [I_{\text{and}}(T) + I_{\text{or}}(T)] + \sum_{T \supseteq (S \cup \{i\}), T \not\ni j} \frac{|S| + 1}{|T|} [I_{\text{and}}(T) + I_{\text{or}}(T)] \right) \\
& - \left( \sum_{T \supseteq (S \cup \{i,j\})} \frac{|S| + 1}{|T|} [I_{\text{and}}(T) + I_{\text{or}}(T)] + \sum_{T \supseteq (S \cup \{j\}), T \not\ni i} \frac{|S| + 1}{|T|} [I_{\text{and}}(T) + I_{\text{or}}(T)] \right) \\
=& \sum_{T \supseteq (S \cup \{i\}), T \not\ni j} \frac{|S| + 1}{|T|} [I_{\text{and}}(T) + I_{\text{or}}(T)] - \sum_{T \supseteq (S \cup \{j\}), T \not\ni i} \frac{|S| + 1}{|T|} [I_{\text{and}}(T) + I_{\text{or}}(T)] \\
=& \sum_{T \supseteq S, T \subseteq N \setminus \{i,j\}} \frac{|S| + 1}{|T \cup \{i\}|} [I_{\text{and}}(T \cup \{i\}) + I_{\text{or}}(T \cup \{i\})] - \sum_{T \supseteq S, T \subseteq N \setminus \{i,j\}} \frac{|S| + 1}{|T \cup \{j\}|} [I_{\text{and}}(T \cup \{j\}) + I_{\text{or}}(T \cup \{j\})] \\
=& \sum_{T \supseteq S, T \subseteq N \setminus \{i,j\}} \frac{|S| + 1}{|T| + 1} [(I_{\text{and}}(T \cup \{i\}) - I_{\text{and}}(T \cup \{j\})) + (I_{\text{or}}(T \cup \{i\}) - I_{\text{or}}(T \cup \{j\}))] = 0
\end{aligned}
$$

*i.e.,* $\varphi(S \cup \{i\}) = \varphi(S \cup \{j\})$

Therefore, we prove the Symmetry axiom-$\alpha$ axiom. □

### G.3. Proof of Symmetry axiom-$\beta$

*Proof.* Without loss of generality, we assume $S \cap T = \emptyset$.

According to the definition of the AND/OR interaction, we can get:$\forall K \supseteq L, K \subseteq N \setminus (S \cup T), \forall T' \subseteq T, T' \neq T, \forall S' \subseteq S, S' \neq S, |S'| = |T'|$,

$$I_{\text{and}}(K \cup S \cup T') = \sum_{J \subseteq (K \cup S \cup T')} (-1)^{|K \cup S \cup T'| - |J|} v_{\text{and}}(J)$$

$$I_{\text{or}}(K \cup S \cup T') = -\sum_{J \subseteq (K \cup S \cup T')} (-1)^{|K \cup S \cup T'| - |L|} v_{\text{or}}(N \setminus J)$$

$$I_{\text{and}}(K \cup S' \cup T) = \sum_{J \subseteq (K \cup S' \cup T)} (-1)^{|K \cup S' \cup T| - |J|} v_{\text{and}}(J)$$

$$I_{\text{or}}(K \cup S' \cup T) = -\sum_{J \subseteq (K \cup S' \cup T)} (-1)^{|K \cup S' \cup T| - |L|} v_{\text{or}}(N \setminus J)$$

Then, we have:

$$
\begin{aligned}
&I_{\text{and}}(K \cup S \cup T') - I_{\text{and}}(K \cup S' \cup T) \\
=&\sum_{J \subseteq (K \cup S \cup T')} (-1)^{|K \cup S \cup T'| - |J|} v_{\text{and}}(J) - \sum_{J \subseteq (K \cup S' \cup T)} (-1)^{|K \cup S' \cup T| - |J|} v_{\text{and}}(J) \\
=&\sum_{J \subseteq (K \cup S' \cup T')} \left( \sum_{\substack{A \in S \setminus S' \\ A \neq S \setminus S'}} (-1)^{|K \cup S \cup T'| - |J \cup A|} v_{\text{and}}(J \cup A) - \sum_{\substack{B \in T \setminus T' \\ B \neq T \setminus T'}} (-1)^{|K \cup S' \cup T| - |J \cup B|} v_{\text{and}}(J \cup B) \right) \\
=&\sum_{J \subseteq (K \cup S' \cup T')} \sum_{\substack{A \in S \setminus S', A \neq S \setminus S' \\ B \in T \setminus T', B \neq T \setminus T' \\ |A| = |B|}} (-1)^{|K \cup S' \cup T| - |J \cup B|} [v_{\text{and}}(J \cup A) - v_{\text{and}}(J \cup B)] \\
=&\, 0
\end{aligned}
$$

$$
\begin{aligned}
&I_{\text{or}}(K \cup S \cup T') - I_{\text{or}}(K \cup S' \cup T) \\
=&-\sum_{J \subseteq (K \cup S \cup T')} (-1)^{|K \cup S \cup T'| - |L|} v_{\text{or}}(N \setminus J) + \sum_{J \subseteq (K \cup S' \cup T)} (-1)^{|K \cup S' \cup T| - |L|} v_{\text{or}}(N \setminus J) \\
=&\sum_{J \subseteq (K \cup S' \cup T')} \left( \sum_{\substack{B \in T \setminus T' \\ B \neq T \setminus T'}} (-1)^{|K \cup S' \cup T| - |J \cup B|} v_{\text{or}}(N \setminus (J \cup B)) - \sum_{\substack{A \in S \setminus S' \\ A \neq S \setminus S'}} (-1)^{|K \cup S \cup T'| - |J \cup A|} v_{\text{or}}(N \setminus (J \cup A)) \right) \\
=&\sum_{J \subseteq (K \cup S' \cup T')} \sum_{\substack{A \in S \setminus S', A \neq S \setminus S' \\ B \in T \setminus T', B \neq T \setminus T' \\ |A| = |B|}} (-1)^{|K \cup S' \cup T| - |J \cup B|} [v_{\text{or}}(N \setminus (J \cup B)) - v_{\text{or}}(N \setminus (J \cup A))] \\
=&\sum_{J \subseteq (K \cup S' \cup T')} \sum_{\substack{A \in S \setminus S', A \neq S \setminus S' \\ B \in T \setminus T', B \neq T \setminus T' \\ |A| = |B|}} (-1)^{|K \cup S' \cup T| - |J \cup B|} [v_{\text{or}}((N \setminus (J \cup A \cup B)) \cup A) - v_{\text{or}}((N \setminus (J \cup A \cup B)) \cup B)] \\
=&\sum_{J \subseteq (K \cup S' \cup T')} \sum_{\substack{A \in S \setminus S', A \neq S \setminus S' \\ B \in T \setminus T', B \neq T \setminus T' \\ |A| = |B|}} 0 = 0
\end{aligned}
$$

Besides, according to Equation (6) in the paper, we have: $\varphi(L \cup S) = \sum_{K \supseteq (L \cup S)} \frac{|L \cup S|}{|K|} [I_{\text{and}}(K) + I_{\text{or}}(K)]$ and $\varphi(L \cup T) = \sum_{K \supseteq (L \cup T)} \frac{|L \cup T|}{|K|} [I_{\text{and}}(K) + I_{\text{or}}(K)]$.

Then, we have:

$$\varphi(L \cup S) - \varphi(L \cup T)$$

$$= \sum_{K \supseteq (L \cup S)} \frac{|L \cup S|}{|K|} [I_{\text{and}}(K) + I_{\text{or}}(K)] - \sum_{K \supseteq (L \cup T)} \frac{|L \cup T|}{|K|} [I_{\text{and}}(K) + I_{\text{or}}(K)]$$

$$= \left( \sum_{K \supseteq (L \cup S \cup T)} \frac{|L \cup S|}{|K|} [I_{\text{and}}(K) + I_{\text{or}}(K)] + \sum_{K \supseteq (L \cup S), K \not\supseteq T} \frac{|L \cup S|}{|K|} [I_{\text{and}}(K) + I_{\text{or}}(K)] \right)$$

$$- \left( \sum_{K \supseteq (L \cup S \cup T)} \frac{|L \cup T|}{|K|} [I_{\text{and}}(K) + I_{\text{or}}(K)] + \sum_{K \supseteq (L \cup T), K \not\supseteq S} \frac{|L \cup T|}{|K|} [I_{\text{and}}(K) + I_{\text{or}}(K)] \right)$$

$$= \sum_{K \supseteq (L \cup S), K \not\supseteq T} \frac{|L \cup S|}{|K|} [I_{\text{and}}(K) + I_{\text{or}}(K)] - \sum_{K \supseteq (L \cup T), K \not\supseteq S} \frac{|L \cup T|}{|K|} [I_{\text{and}}(K) + I_{\text{or}}(K)]$$

$$= \sum_{K \supseteq L, K \subseteq N \setminus (S \cup T)} \sum_{T' \subseteq T, T' \neq T} \frac{|L \cup S|}{|K \cup S \cup T'|} [I_{\text{and}}(K \cup S \cup T') + I_{\text{or}}(K \cup S \cup T')]$$

$$- \sum_{K \supseteq L, K \subseteq N \setminus (S \cup T)} \sum_{S' \subseteq S, S' \neq S} \frac{|L \cup T|}{|K \cup S' \cup T|} [I_{\text{and}}(K \cup S' \cup T) + I_{\text{or}}(K \cup S' \cup T)]$$

$$= \sum_{\substack{K \supseteq L \\ K \subseteq N \setminus (S \cup T)}} \sum_{\substack{T' \subseteq T, T' \neq T \\ S' \subseteq S, S' \neq S \\ |S'| = |T'|}} \frac{|L \cup T|}{|K \cup S' \cup T|} [I_{\text{and}}(K \cup S \cup T') - I_{\text{and}}(K \cup S' \cup T) + I_{\text{or}}(K \cup S \cup T') - I_{\text{or}}(K \cup S' \cup T)]$$

$$= 0$$

Therefore, we prove the Symmetry axiom-$\beta$ axiom. $\qquad\square$

### G.4. Proof of Additivity axiom

*Proof.* According to the definition of the AND/OR interaction, we can get:

$I_{\text{and}_v}(T) = \sum_{L \subseteq T} (-1)^{|T|-|L|} v_{\text{and}}(L), \ I_{\text{or}_v}(T) = -\sum_{L \subseteq T} (-1)^{|T|-|L|} v_{\text{or}}(N \setminus L)$

$I_{\text{and}_{v_1}}(T) = \sum_{L \subseteq T} (-1)^{|T|-|L|} v_{\text{and}_1}(L), \ I_{\text{or}_{v_1}}(T) = -\sum_{L \subseteq T} (-1)^{|T|-|L|} v_{\text{or}_1}(N \setminus L)$

$I_{\text{and}_{v_2}}(T) = \sum_{L \subseteq T} (-1)^{|T|-|L|} v_{\text{and}_2}(L), \ I_{\text{or}_{v_2}}(T) = -\sum_{L \subseteq T} (-1)^{|T|-|L|} v_{\text{or}_2}(N \setminus L)$

where $v(L) = v_{\text{and}}(L) + v_{\text{or}}(L)$, $v_1(L) = v_{\text{and}_1}(L) + v_{\text{or}_1}(L)$ and $v_2(L) = v_{\text{and}_2}(L) + v_{\text{or}_2}(L)$.

Due to $v(L) = v_1(L) + v_2(L)$, we have: $I_{\text{and}_v}(T) = I_{\text{and}_{v_1}}(T) + I_{\text{and}_{v_2}}(T)$ and $I_{\text{or}_v}(T) = I_{\text{or}_{v_1}}(T) + I_{\text{or}_{v_2}}(T)$.

According to Equation (6) in the paper, we have: $\varphi_v(S) = \sum_{T \supseteq S} \frac{|S|}{|T|} [I_{\text{and}_v}(T) + I_{\text{or}_v}(T)]$, $\varphi_{v_1}(S) = \sum_{T \supseteq S} \frac{|S|}{|T|} [I_{\text{and}_{v_1}}(T) + I_{\text{or}_{v_1}}(T)]$, $\varphi_{v_2}(S) = \sum_{T \supseteq S} \frac{|S|}{|T|} [I_{\text{and}_{v_2}}(T) + I_{\text{or}_{v_2}}(T)]$

Then, we have:

$$\varphi_v(S) = \sum_{T \supseteq S} \frac{|S|}{|T|} [I_{\text{and}_v}(T) + I_{\text{or}_v}(T)]$$

$$= \sum_{T \supseteq S} \frac{|S|}{|T|} \left[ (I_{\text{and}_{v_1}}(T) + I_{\text{and}_{v_2}}(T)) + (I_{\text{or}_{v_1}}(T) + I_{\text{or}_{v_2}}(T)) \right]$$

$$= \sum_{T \supseteq S} \frac{|S|}{|T|} [I_{\text{and}_{v_1}}(T) + I_{\text{or}_{v_1}}(T)] + \sum_{T \supseteq S} \frac{|S|}{|T|} [I_{\text{and}_{v_2}}(T) + I_{\text{or}_{v_2}}(T)]$$

$$= \varphi_{v_1}(S) + \varphi_{v_2}(S)$$

Therefore, we prove the Additivity axiom. $\qquad\square$

### G.5. Proof of Dummy axiom

*Proof.* According to the definition of the AND/OR interaction, we can get:

$$\forall T \ni i, I_{\text{and}}(T) = \sum_{L \subseteq T}(-1)^{|T|-|L|}v_{\text{and}}(T) = \sum_{L \subseteq T \setminus \{i\}}(-1)^{|T|-|L|+1}\left[v_{\text{and}}(L \cup \{i\}) - v_{\text{and}}(L)\right]$$

$$\forall T \ni i, I_{\text{or}}(T) = -\sum_{L \subseteq T}(-1)^{|T|-|L|}v_{\text{or}}(N \setminus L) = \sum_{L \subseteq T \setminus \{i\}}(-1)^{|T|-|L|+1}\left[v_{\text{or}}(N \setminus L) - v_{\text{or}}(N - L - \{i\})\right]$$

Due to $\forall T \subseteq N \setminus \{i\}, v(T \cup \{i\}) = v(T)$, we have: $\forall T \subseteq N \setminus \{i\}, v_{\text{and}}(T \cup \{i\}) = v_{\text{and}}(T), v_{\text{or}}(T \cup \{i\}) = v_{\text{or}}(T)$.

Thus, we have:

$$I_{\text{and}}(T) = \sum_{L \subseteq T \setminus \{i\}}(-1)^{|T|-|L|+1}\left[v_{\text{and}}(L \cup \{i\}) - v_{\text{and}}(L)\right] = 0$$

$$I_{\text{or}}(T) = \sum_{L \subseteq T \setminus \{i\}}(-1)^{|T|-|L|+1}\left[v_{\text{or}}(N \setminus L) - v_{\text{or}}(N - L - \{i\})\right] = 0$$

Then, according to Equation (6) in the paper, we get:

$$\varphi(S) = \sum_{T \supseteq S}\frac{|S|}{|T|}\left[I_{\text{and}}(T) + I_{\text{or}}(T)\right] = \sum_{T \supseteq S, T \ni i}\frac{|S|}{|T|}\left[I_{\text{and}}(T) + I_{\text{or}}(T)\right] = 0$$

Therefore, we prove the Dummy axiom. $\square$

### G.6. Proof of Corollary 8

*Proof.* According to the Efficiency axiom of the Shapley value, we have: $v(N) - v(\emptyset) = \sum_{i \in N}\phi(i)$.

Then, according to Theorem 4, we have: $\sum_{i \in S}\phi(i) = \varphi(S) + \sum_{T \subseteq N, T \cap S \neq \emptyset, T \cap S \neq S}\frac{|T \cap S|}{|T|}\left[I_{\text{and}}(T) + I_{\text{or}}(T)\right]$.

Thus, we have:

$$
\begin{aligned}
v(N) - v(\emptyset) &= \sum_{i \in N}\phi(i) \\
&= \sum_{i \in S}\phi(i) + \sum_{i \in N \setminus S}\phi(i) \\
&= \varphi(S) + \sum_{i \in N \setminus S}\phi(i) + \sum_{T \subseteq N, T \cap S \neq \emptyset, T \cap S \neq S}\frac{|T \cap S|}{|T|}\left[I_{\text{and}}(T) + I_{\text{or}}(T)\right]
\end{aligned}
$$

Therefore, we prove Corollary 8. $\square$

## H. Detailed introduction about the axioms for the coalition attribution

The anonymity axiom shows that the order of input variables does not essentially affect the coalition's attribution $\varphi(S)$. The symmetry axiom shows that if two coalitions always have the same roles, then they have exactly the same attributions. The additivity axiom shows if the model output $v(S)$ can be represented as the sum of outputs of two sub-models $v(S) = v_1(S) + v_2(S)$, then a coalition's attribution can be decomposed into attributions computed on the two sub-models. The dummy axiom shows that if the coalition $S$ contains a dummy input variable $i$, which does not contribute to the model output, then the coalition S has zero attribution (but the coalition $S \setminus \{i\}$ may have non-zero attribution $\varphi(S \setminus \{i\})$).

## I. Results of coalition faithfulness metrics on the image data

We evaluated whether these DNNs accurately represented natural coalitions in human cognition. We trained VGG-11 (Simonyan & Zisserman, 2014) and ResNet-20 (He et al., 2016) on the MNIST (LeCun et al., 1998) and CIFAR-10 (Krizhevsky, 2012) datasets. We divided an image sample into $8 \times 8$ regions and manually selected 10 image regions, which included some neighboring regions with specific semantics and some other random regions. The set of regions that represented clear visual concepts was annotated as a true coalition. In comparison, a random set of image regions was annotated as a false coalition.

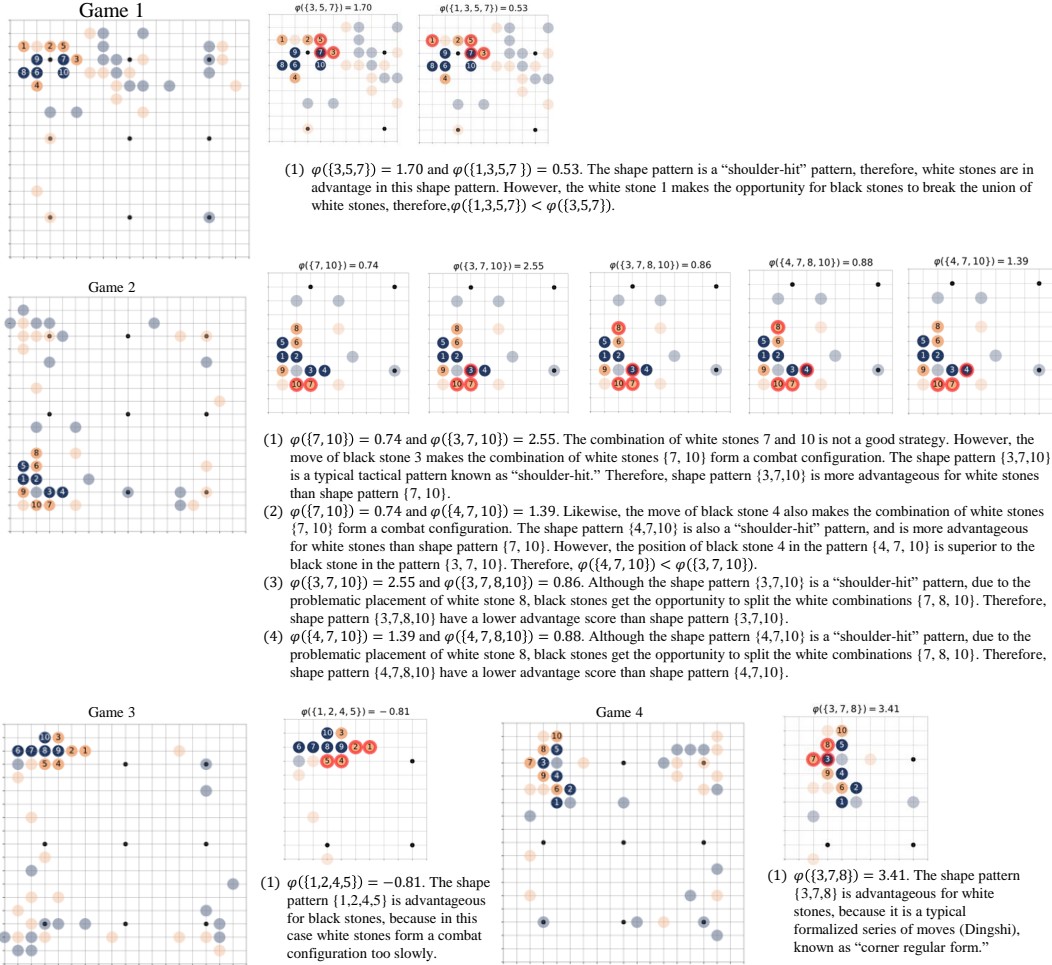

*Figure 3.* Analysis of shape patterns in Go compared to human intuition

Figure 4 and 5 respectively show the results of coalition attribution faithfulness metrics of VGG-11 and ResNet-20 model on the CIFAR-10 dataset. Figure 6 and 7 respectively show the results of coalition attribution faithfulness metrics of VGG-11 and ResNet-20 model on the MNIST dataset. In these figures, $x_1, ..., x_{10}$ represent the selected image regions, and the image regions marked in yellow represent the selected coalition $S$. These results show that the manually selected coalitions, like $\{x_0, x_1, x_4\}$ on the left side of the top row in Figure 4, which represents the head of the horse, had high $Q(S), R(i), R'(i)$ values and were considered as faithful coalitions. In comparison, the randomly selected coalitions with low high $Q(S), R(i), R'(i)$ values, were considered as unfaithful coalitions.

## J. Comparison with human intuitions in Go game

This experiment applies our theoretical framework to uncover shape patterns implicitly learned by the network, many of which go beyond traditional human knowledge.

We hired 5 expert Go players to analyze the fitness between the extracted coalitions and human intuition on many more game boards. Under their guidance, we focused on explaining shape patterns involving a limited number of stones, as determined by the experts. Specifically, we excluded coalitions with more than six stones, as such complex configurations were often found to have a negligible impact and likely reflect noise. Figure J shows the detailed analysis of shape patterns in Go compared to human intuition. As shown in Figure J, some automatically learned coalitions do not fit human intuition. As a possible explanation for this, human players typically assess patterns based on short-term tactical search and a few-step lookahead, but the value network implicitly captures long-term statistical regularities from many more games. Although these long-term patterns are difficult to interpret directly, they may reveal new shape patterns. Expert Go players say they have learned some new knowledge from these patterns.

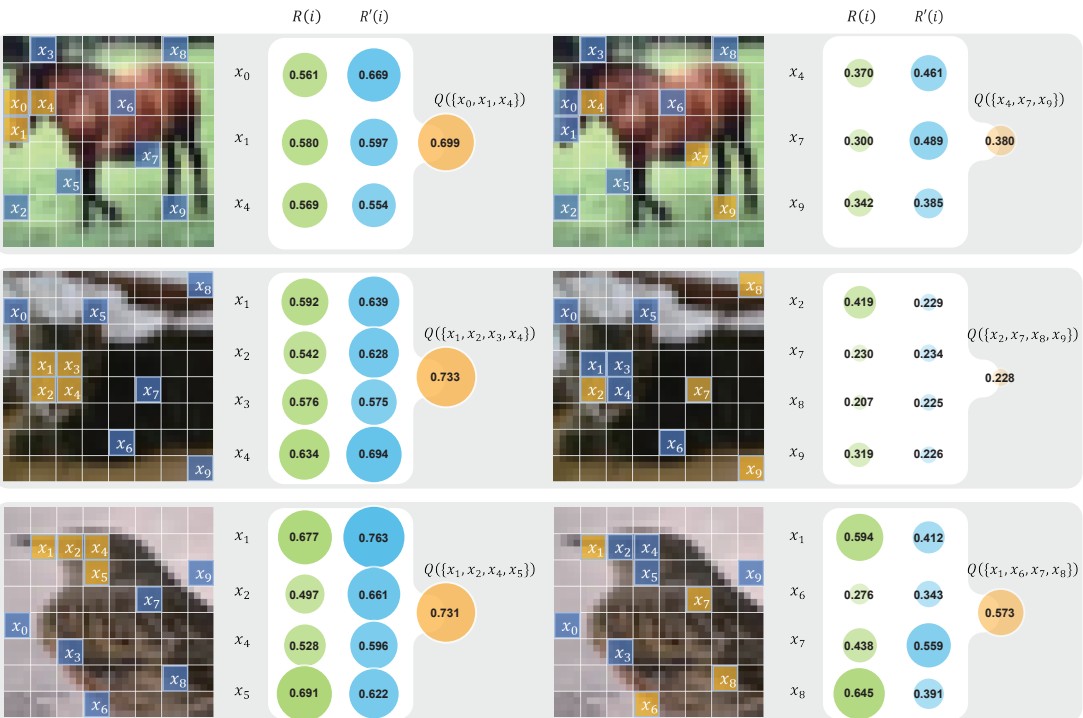

Figure 4. Coalition attribution faithfulness metrics of VGG-11 on CIFAR-10 dataset

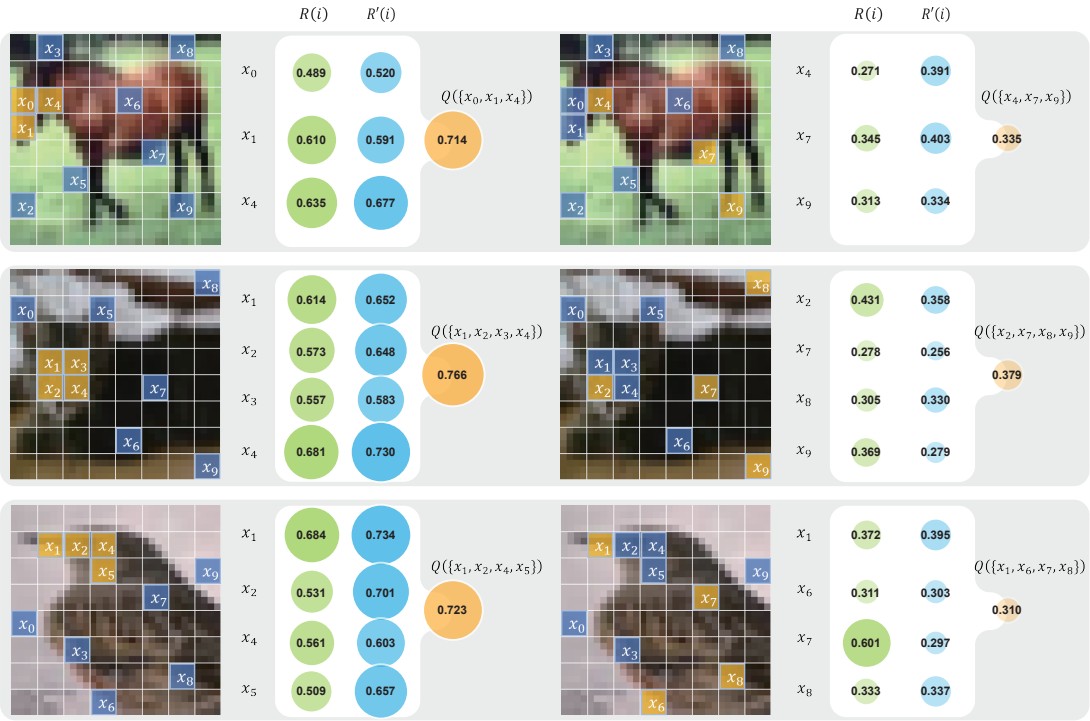

Figure 5. Coalition attribution faithfulness metrics of ResNet-20 on CIFAR-10 dataset

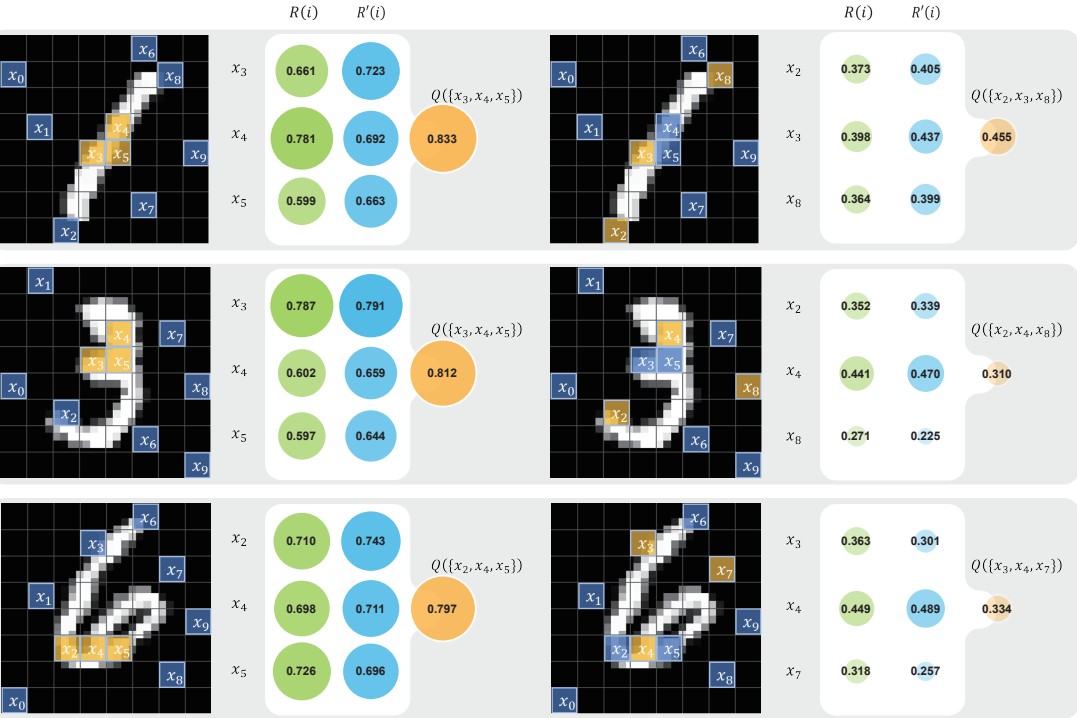

*Figure 6.* Coalition attribution faithfulness metrics of VGG-11 on MNIST dataset

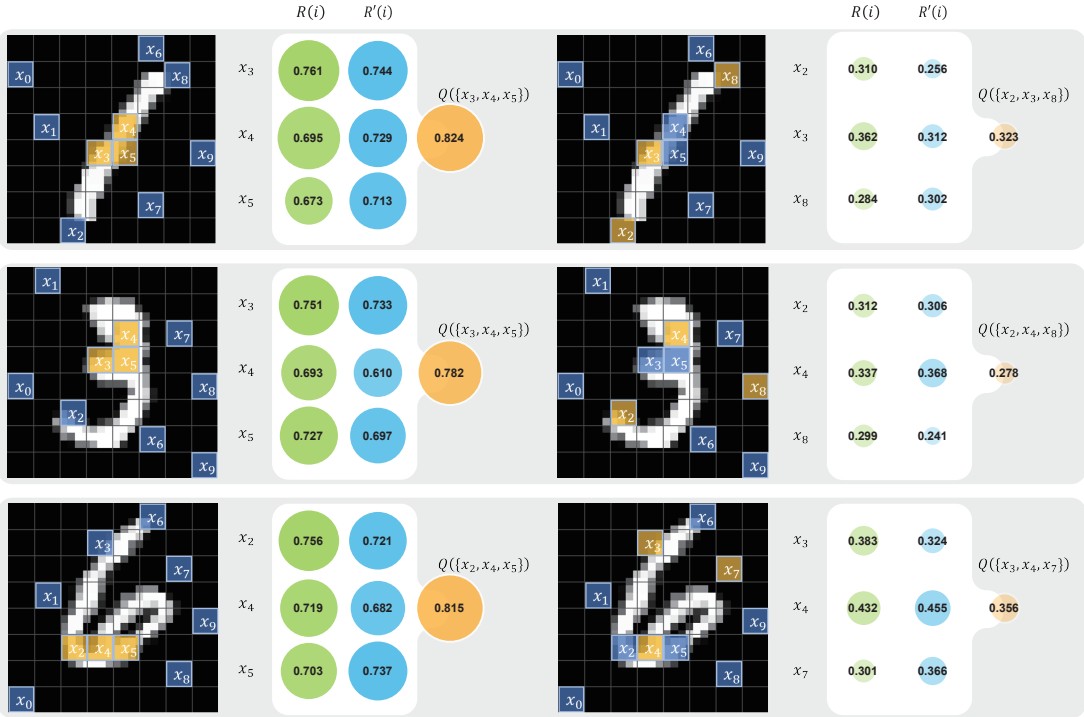

*Figure 7.* Coalition attribution faithfulness metrics of ResNet-20 on MNIST dataset

