# OpenReview forum: "Towards Attributions of Input Variables in a Coalition"
_ICML.cc/2025/Conference — ICML 2025 poster_

### Official Review · Reviewer_mhJ5 · 2025-02-27

**Overall Recommendation:** 3

**Summary:**

This paper studies the partitioning of input variables in feature attribution methods. The central issue is that existing attribution methods compute importance scores of single features or predefined partitions, but they are not very good at attribution for meaningful coalitions of variables. The paper identifies fundamental conflicts in coalition attributions and provides a new method with theoretical guarantees. In particular, it analyzes AND-OR interactions ti reveal how feature interactions impact attributions and extends the Shapley value to define a new coalition attribution metric that accounts for interactions among variables. This method is validated on synthetic functions, NLP, image classification, and the game of Go, demonstrating consistency with human intuition.

## update after rebuttal
The rebuttal addressed my concerns, so I kept my positive score.

**Claims And Evidence:**

Yes.

- Thoeretical part: the paper reformulates the Shapley and Banzhaf values in terms of AND-OR interactions (Theorems 3.2 and 3.3). The new coalition attribution method is then shown to be consistent with these formulations.

- Empirical part: Three proposed faithfulness metrics (R(i),  R'(i), and Q(S)) are introduced to effectively measure the validity of a coalition. They are first verified on models fitting synthetic functions and then experiments on NLP tasks, image classification, and Go game.

**Essential References Not Discussed:**

The coverage of related work is reasonable, for feature interaction explanations. The XAI literature is too huge to cover comprehensively.

**Experimental Designs Or Analyses:**

Yes, the experiment design is reasonable, including NLP tasks, image classification, and the Go game. One issue might be these evaluations are more like case studies, where only certain coalitions can be covered, so the generalization is completely guaranteed.

**Methods And Evaluation Criteria:**

Yes. The proposed metric makes sense and the evaluation is relatively thorough.

**Other Comments Or Suggestions:**

None.

**Other Strengths And Weaknesses:**

Strengths
 - The paper has a strong theoretical grounding. It provides a rigorous mathematical explanation for attribution conflicts.
 - The proposed method applies to multiple real-world tasks (NLP, images, and Go), and has been verified on all of them.
 - The AND-OR interaction framework offers an intuitive way to understand feature interactions, which I really like.

Weaknesses
 - No time complexity or scalability analysis. The experiments involve small coalitions (≤10 variables). It is unclear whether the method scales to larger feature sets, which is a big concern for feature-interaction explanation.
 - Although the Go application is interesting, it is unclear how about the direct use case of this method in NLP and image domains. My understanding is that users always need to predefine coalitions rather than having the model identify the top important ones.

**Questions For Authors:**

- Time Complexity or Scalability: What is the computational complexity of the method? How does the method perform when applied to high-dimensional feature spaces?

- What are some concrete applications of NLP and Vision?

**Relation To Broader Scientific Literature:**

The paper is well-situated within the explainable AI (XAI) and feature interaction literature. It builds upon: Shapley value-based attributions, feature interaction explanations, and general game-theoretic approaches.

**Theoretical Claims:**

Yes, theoretical claims are provided in App Appendix. I scanned through the the proof of Thm 2 and 3 in the Appendix. Seems to be correct.

---

> ### Author Rebuttal · Authors · 2025-04-01
>
> Thank you very much for your great efforts in reviewing this paper. We would like to answer all your concerns. **Please let us know if you have further questions or if you are not satisfied with the current responses.**
>
> **Q1: “No time complexity or scalability analysis.”**
> > The experiments involve small coalitions … feature-interaction explanation.
>
> A: A good question. We have followed your suggestions to provide a comprehensive analysis of the time complexity and scalability of our method.
>
> **Computational complexity:** First, the computational complexity of computing a coalition’s attribution is $O(2^n)$, which is the same as the computational complexity of the Shapley value and the Banzhaf value. Second, the computational complexity of computing AND-OR interactions is $O(2^n)$.
>
> **Scalability:** The large computational complexity is a common issue for many attribution methods (e.g., Shapley value and Banzhaf value) to scale up. To this end, we can apply the fast approximation sampling method proposed by Kang et al. at UC Berkeley [cite 1], which is the only method to speed up the extraction of AND interactions, to the best of our knowledge. Extending this method to extract AND-OR interaction is the next issue in future work. Nevertheless, this fast approximation sampling strategy provides a new hope to speed up the computation of the attribution. We will discuss in the revised paper.
>
> As another widely-used strategy for scalability, previous studies[cite 2, cite 3 cite 4] usually define each input variable as a larger image or a longer phrase in a sentence to reduce the number of input variables, thereby reducing the computational cost. However, the attribution conflict problem is usually more serious when we use larger but less input variables. To this end, our theory naturally discovers the interactions that cause the attribution conflict, which help people understand the coalition attribution.
>
> [cite 1] Justin S. Kang, Yigit E. Erginbas, Landon Butler, Ramtin Pedarsani, Kannan Ramchandran “Learning to Understand: Identifying Interactions via the Möbius Transform”
>
> [cite 2] Ren et al. “Defining and quantifying the emergence of sparse concepts in DNNs”
>
> [cite 3] Li et al. “Does a neural network really encode symbolic concepts?”
>
> [cite 4] Ren et al. “Where we have arrived in proving the emergence of sparse symbolic concepts in AI models”
>
>
> ---
>
> **Q2: It is unclear about the direct use case of this method in NLP and image domains. My understanding is that users always need to predefine coalitions rather than having the model identify the top important ones.**
> **What are some concrete applications of NLP and Vision?**
>
> A: A good question. Because our theory clarifies in mathematics the underlying cause for the attribution conflict problem ubiquitously appearing in different attribution methods, we can simply use our theory to automatically distinguish faithful coalitions and unfaithful coalitions.
>
> Specifically, we can extract all AND-OR interactions between different elementary input variables. Because the attribution conflict w.r.t. the coalition $S$ is caused by numerical effects of all interactions $T$ that contain just partial but not all variables in $S$, we can identify faithful coalitions, i.e., most input variables in a faithful coalition $S$ are supposed to appear together in different interactions, instead of appearing separately. Therefore, our theory provides an essential perspective to identify natural coalitions automatically learned by an NLP model or a vision model.
>
> In other words, the mining of faithful coalitions can be formulated as the discovery of common subgroups in the extracted interactions. In particular, the proven sparsity property of AND-OR interactions significantly reduces the computational cost of mining faithful coalitions.
>
> Besides, our theory can also evaluate the representation quality of a DNN. Given a set of faithful coalitions (with large values of $Q(S),R(i),R’(i)$) discovered by our theory, we can examine whether these coalitions encode interaction between obviously irrelevant input variables. If so, it can be considered as a representation flaw of a DNN. For example, if a coalition contains both foreground image patches and background image patches, or if a coalition contains both tokens related to the generated language and tokens irrelevant to the generated language, then this coalition usually represents a representation flaw.

---

> > ### Comment · Reviewer_mhJ5 · 2025-04-02
> >
> > Thank the authors for their response. I will keep my score.

---

> > > ### Author Response · Authors · 2025-04-05
> > >
> > > Thank you very much for your great efforts in reviewing this paper.

---

### Official Review · Reviewer_ZrgL · 2025-03-10

**Overall Recommendation:** 1

**Summary:**

This paper proposes a new perspective on common attribution methods such as Shapely values and Banzhaf values. The paper does so in a quite theoretical way illustrating that one can reformulate the computation of these attribution methods in terms of "AND" and "OR" interactions. The AND interactions are "hot" when all players of this interaction are present in a coalition and the OR interaction is "hot" when at least one player of this interaction is present in a coalition. Once an interaction is "hot" it contributes to the worth of a coalition (is part of the value functions output).

## update after rebuttal
The rebuttal did not alleviate my concerns with the paper. I still like the theoretical work this paper is doing and the novel representation decomposing the value function into I_and and I_or interactions.

- I still see issues with this work's Go experiment. I do not see how this work's "explanations" or better attributions **(a)** help expert Go players (as the work claims but _does not provide any evidence for_) or **(b)** shows how only with the novel representation we can come to new insights we otherwise could not have come to by using other explanation methods (which might be flawed but still get the job done).
- Likewise, I still think that this method is not well compared and put into context with the current stream of attribution methods. The criticisms towards the current state-of-the-art and the papers novel interpretation of interactions are not compared to the current explanation literature. The paper is spending a lot of time on arguing that current attribution methods are flawed but **does not provide enough evidence where current attribution methods fail or how big this problem practically is**.
- After checking the replies by the authors, I was left with more questions regarding the empirical evaluation and wanted to check this work's implementation. However, the authors **do not provide any code** for review.

**Claims And Evidence:**

The main contribution of this work is that it illustrates that the worth of coalition can be represented with AND and OR interactions. It summarizes the current stream of literature about Shapley Interactions and shows that all contribution methods to some extend are not so intuitive. This in my opinion is a nice perspective and research question to take. The paper contains many examples (some more and some less easy to follow) showing where their perspective makes sense and is reasonable. The paper contains a few theoretical results.

However, the experimental evaluation, does not really help and advocate why the new perspective on attribution methods is important for the general machine learning community. Yes, Shapley values and Banzhaf values are important for many machine learning settings. However, this paper spends a lot of time on the game "Go" and very little on machine learning models. After reading the paper and especially the empirical sections, I still do not understand how the new representation actually helps feature attribution, data valuation or other important application settings of the Shapley/Banzhaf value. Yet, this would be this paper's most important job.

**Essential References Not Discussed:**

Seems fine.

**Experimental Designs Or Analyses:**

I checked the experiments, but do not see how they should convince the general machine learning community to make use of the new representation of the attribution methods presented here.

**Methods And Evaluation Criteria:**

I do not understand how the methods evaluation criteria helps the general machine learning community to make use of the new representation of the attribution methods presented here.

**Other Comments Or Suggestions:**

The paper is actually very interesting. However, in its current form it is very hard to follow and does not present a clear and convincing argument for your method. I strongly suggest to take a step back and revise the paper from the beginning. There are way too many examples with toy value functions and data modalities across the sections (starting off with vision + synthetic interactions in Fig. 1, then the natural text examples with the sentiment analysis, then the "raining cats and dogs examples" then the game of Go). This is just too much and takes away from your actual contribution. I am not saying you should forfeit all these examples. It is good to show that your method is generally applicable to a wide range of machine learning scenarios, but as it stands right now it is just all over the place and needs to be streamlined.

I recommend to start of with one motivating example in the beginning and then taking a more abstract view throughout your methodology and theoretical background section, such that you can expand again in the experimental section at the end. There, clear examples from different ML domains are very much appreciated.

__Sidenote__: Use less boldface and repetition in the sentences.

**Other Strengths And Weaknesses:**

### Strengths
- I like the perspective of the paper and the theoretical work and I think it studies a very nice research problem.

### Weaknesses
- I think that the paper is from a methodology standpoint quite limited as it stands right now. The work should focus much more on the new representation of these attribution methods and use them for something meaningful like either a) computing attribution values more efficiently or b) creating more faithful explanations for machine learning applications.
- I find this paper very hard to read and follow. It took me quite some time to get the gist of it (more on that in other comments or suggestions).

**Questions For Authors:**

- Can we disentangle the AND and OR interactions from each other in some methodologically novel way? Currently the paper just argues for this new representation, but can this also be used either a) more efficiently compute attribution values or b) be used to more efficiently _explain_ machine learning models with different kinds of attribution scores?

**Relation To Broader Scientific Literature:**

Since this work focuses on game theoretic foundations on how to represent attribution scores, it touches on many machine learning applications. Most predominantly, explainable AI or data valuation. Specifically this work touches on the domain of interaction quantification, which currently is getting more attention.

**Theoretical Claims:**

The theoretical claims of this work are interesting and up-to my knowledge novel. I very much appreciate the perspective this paper takes on Shapley values and their interpretation of different kind of interactions. I checked Theorem 3.2 and Theorem 3.3 in detail. While, the proofs could actually benefit from some comments (especially when sets in summations are getting restructured), I could follow along quite well. They seem to be correct.

---

> ### Author Rebuttal · Authors · 2025-04-01
>
> Thank you for your comments. **Please feel free to contact us if you have any further concerns as soon as possible.**
>
> ---
>
> **Q1: Ask for the significant value of our method. Why not focus on computational efficiency or design new methods?**
> > “I do not understand how … community”
>
> > “I think that the paper is … right now.”
>
> A: A good question. Analyzing and debugging mathematical problems with attribution methods represents a newly emerging research direction of attribution methods[1,5,6]. Compared to boosting computational efficiency or explanation accuracy, *people gradually realize that the lack of a clear explanation for inherent mathematical limitations in attribution methods have hampered the future development of attribution methods, as follows*
>
> $\bullet$ **Background 1:** Most attribution methods are designed empirically, previous explanation theories cannot obtain mutual consistency between them or clarify their theoretical foundations [4].
>
> $\bullet$ **Background 2:** Most evaluation metrics for attributions have been found to have obvious flaws (see discussion in [4]). More crucially, constructing a benchmark model with a ground-truth attribution for evaluation also presents a significant challenge [2,3].
>
> **Therefore, besides designing a new attribution method, debugging the mathematical problem of different attribution methods represents a new challenge for XAI.** [5,6] all attempt to explain or unify the underlying mechanisms of different attribution methods.
>
> To this end, our study explores a new perspective, i.e., explaining the mathematical factor that causes the conflict of attributions. This is one of the most common issues that are shared by almost all attribution methods, but have not been sophisticatedly investigated.
>
> **New experiments to show the commonness of the conflict problem on various DNNs.**
>
> We demonstrate attribution conflicts by comparing 6 attribution methods across BERT-large, LLaMA, and VGG-11. Most attribution methods all exhibit conflicted attributions, but they fail to explain the internal mechanism. In comparison, our attribution method first clarifies a set of interaction effects as the hidden cause for the attribution conflict, and helps people identify representation flaws (incorrect attributions/interactions) in a DNN.
>
> Please see the results (conflict.pdf) in https://gofile.io/d/0azDsA.
>
> ---
>
> **Q2: "However, the experimental evaluation", ... "how the new representation actually helps feature attribution ..."**
>
> A: Thanks. First, our theory of explaining the internal conflict of attributions **does** help us design a new attribution method for coalitions, which first clarifies the intrinsic cause of the conflict problem. Please see Section 3.4 for details.
>
> Second, our theory also provides a new mathematical metric to evaluate the faithfulness of coalitions, because breaking a faithful coalition will cause lots of internal conflicts. Thus, we conduct different experiments to use our theory to evaluate the coalition.
>
> (1) We build up a benchmark to evaluate whether our metrics could effectively evaluate the coalition discovery, i.e., distinguishing the faithfulness of a coalition (see Table 4).
>
> (2) Table 5  in the main text and Figure 3-5 in the Appendix evaluate the faithfulness of the coalitions for both NLP models and vision models. These all provide new insights into the attribution.
>
> (3) The experiment in Figure 2 uses our theory to explain shape patterns used by the DNN to play the Go game. Our method helps expert Go players learn new shape patterns (beyond traditional knowledge of the game) to play the Go game.
>
> ---
>
> **Q3: About paper writing**
>
> A: Thanks a lot. We follow your suggestions to carefully polish the language.
>
> ---
>
> **Q4: Can we disentangle the AND and OR interactions from each other in some methodologically novel way?**
>
> A: The definition and decomposition of AND-OR interactions (see Eq. 3-4) has been a well-established research direction. We have followed the standard methods widely used by [7,8] to extract AND-OR interactions. To this end, Kang et al.[9] at UC Berkeley have developed a more efficient but approximate way to compute AND interactions, but this method can only extract AND interactions.
>
> ---
>
> [1] Kumar et al. “Problems with Shapley-value-based explanations as feature importance measures”
>
> [2] Yang et al. “Benchmarking attribution methods with relative feature importance”
>
> [3] Rao et al. “Towards Better Understanding Attribution Methods”
>
> [4] Deng et al. “Unifying fourteen post-hoc attribution methods with taylor interactions”
>
> [5] Lundberg et al. “A unified approach to interpreting model predictions”
>
> [6] Sixt et al. “When explanations lie: Why many modified bp attributions fail”
>
> [7] Li et al. “Does a Neural Network Really Encode Symbolic Concept?”
>
> [8] Ren et al. “Towards the Dynamics of a DNN Learning Symbolic Interactions“
>
> [9] Kang et al. “Learning to Understand: Identifying Interactions via the Möbius Transform”

---

> > ### Comment · Reviewer_ZrgL · 2025-04-07
> >
> > Dear authors,
> >
> > thank you for your detailed response. Unfortunately, I am still not very convinced of the paper.
> >
> > From a critique perspective, the work still has some problems and does not get its point across:
> > - The additional results you provided do not alleviate this problem. In most of the experiments you are arguing about faithfulness of the explanations towards explaining coalition values. However, you do not evaluate what happens when you use a method designed for _faithfulness_ like Faithful Shapley/Banzhaf regression by Tsai et al (2023) which is absent from your Figure 1. in the additional results contained in the link. No experiments in the paper compare or evaluate how different **state-of-the-art attribution methods deal with or breaks due to this problem**. The additional results in the link is the first time Shapley, Banzhaf, or Interaction Indices are analyzed. This makes it hard for practitioners and researchers alike to gauge the impact of the problem and whether this is actually is a problem worth trying to solve.
> > - _"Our method helps expert Go players learn new shape patterns (beyond traditional knowledge of the game) to play the Go game."_ In the paper, you do not showcase or measure this. How do you now that you method **helps expert Go players**? The whole experiment with the game of Go is very synthetic and limited in nature showing only two board states and limiting on 10 stones (players).
> > - _"Table 5 in the main text and Figure 3-5 in the Appendix evaluate the faithfulness of the coalitions for both NLP models and vision models. These all provide new insights into the attribution." These again, are synthetic and unreliable examples which do not help quantifying the problem on a broader scale (2 Sentences and 6 Images with coalitions chosen by human-intuition).
> >
> > From a **methodological perspective** (as mentioned with _my question for the authors_), the work is still quite limited. The work brings forward a different representation of the Banzahf and Shapley value, but stays abstract. My question was "can we use this representation to make something better and actually compute something better". While I do not say that this work would need to solve the problem in its entirety (identification of a problem is of course also an important contribution), however offering any remedy in that regard would be a start. Specifically since it is quite well known that attribution methods are limited in their explanatory power as you already point out or further analyzed in [1, 2] necessitating explanations of higher-orders (however they may look like). Yes, the method by Kang et al. (2024) does only compute AND interactions. Similarly, do all the interaction methods contained in shapiq [3] or in the vast body of game-theoretic literature on interactions. Having a bad but working baseline from your new and unstudied perspective on interactions would be greatly appreciated.
> >
> > ### Sidenote from viewing Figure 1 in the Addendum:
> > For theoretical and empirical evaluation the Shapley interactions proposed by Bordt and Luxbourg [2] are very handy compared to the Shapley interaction index by Grabisch and Roubens [4], which probably should not be used for feature attribution purposes directly since it is not efficient.
> >
> > ### References:
> > - 1: Tsai et al. (2023) "Faith-Shap: The Faithful Shapley Interaction Index" link: https://jmlr.org/papers/v24/22-0202.html
> > - 2: Bordt and von Luxburg 2023 "From Shapley Values to Generalized Additive Models and back" link: https://proceedings.mlr.press/v206/bordt23a/bordt23a.pdf
> > - 3: Muschalik et al. (2024) "shapiq: Shapley Interactions for Machine Learning" https://proceedings.neurips.cc/paper_files/paper/2024/hash/eb3a9313405e2d4175a5a3cfcd49999b-Abstract-Datasets_and_Benchmarks_Track.html
> > - 4: Grabisch and Roubens "An axiomatic approach to the concept of interaction among players in cooperative games" https://link.springer.com/article/10.1007/s001820050125
> >
> > Kind Regards, Reviewer ZrgL

---

> > > ### Author Response · Authors · 2025-04-09
> > >
> > > Thanks a lot. Due to the limited time window of < 48 hours after your reply, we are pleased to conduct new experiments to answer your new concerns. All results will be put on the paper.
> > >
> > > ---
> > >
> > > **Q1: Ask for new experimental results. “Do not evaluate ... like Faithful Shapley/Banzhaf regression by Tsai et al (2023) ”**
> > >
> > > A: Thank you for your comment. As the supplementary to the last reply, we are pleased to add experiment results of attribution conflicts generated by the method [Tsai et al (2023)] in the following link: https://anonymous.4open.science/r/ICML_rebuttal-6D88/FaithShap.pdf.
> > >
> > > ---
> > >
> > > **Q2: Ask about "how different state-of-the-art attribution methods deal with or breaks due to this problem." "This makes it hard for practitioners and researchers alike to gauge the impact of the problem and whether this is actually is a problem worth trying to solve."**
> > >
> > > A: A good question. First, we need to clarify that our study just explains the mechanism that causes the conflict and proposes a method with transparent mechanisms, **so theoretically, our research cannot be directly compared with the methods of alleviating the conflict.**
> > >
> > > Despite that, we would like to extend our theory to these methods. (1) Table 1 has introduced all previous attempts, which partially solve the conflict problem, in which Shapley value and Banzhaf value are representives. (2) As a theoretical foreshadowing, Theorems 3.2 and 3.3 have further prove that both attributions can be formulated in our paradigm of interaction allocation.
> > >
> > > **Banzhaf value.** In this way, our theory can be simply extended to the Banzhaf value.
> > >
> > > **Theorem 1**:  Similar to the Shapley value, the attribution of a coalition $S$ can be formalized as $\varphi_{B}(S) = \sum_{T\supseteq S}\frac{1}{2^{|T|-|S|}}\left[I_{\text{and}}(T)+I_{\text{or}}(T)\right]$. Then, the attribution conflict $B_{\text{conflict}}(S) \overset{\text{def}}{=} \sum_{i\in S} B(i)- B_{\text{shared}}(S)$, subject to $B_{\text{shared}}(S)\overset{\text{def}}{=}\varphi_B(S)$, can be explained as follows:
> > >
> > > $B_{\text{conflict}}(S)=\sum_{T\subseteq N, T\cap S \neq \emptyset, T\cap S \neq S}{\frac{1}{2^{|T \setminus S|}}}\left[I_{\text{and}}(T)+I_{\text{or}}(T)\right]$
> > >
> > > **Faith-Shap.** defines a family of relative faithful interaction indices. We have also tested the conflict on the Faith-Shap in **new experiments**. Please see answers to Q1.
> > >
> > > ---
> > >
> > > **Q3 Ask for comparison with human intuition. "How do you now that your method helps expert Go players?” “on a broader scale … by human intuition)”**
> > >
> > > A: We are pleased to add the new comparison with human intuitions. We have conducted new experiments. We have hired 5 expert Go players to analyze the fitness between the extracted coalitions and human intuition on much more game boards.
> > >
> > > Following your suggestions, we choose to explain more shape patterns under the guidance of expert Go players. The number of stones is determined by these experts. Following the guidance of experts, we do not test coalitions with more than 6 stones, because too complex coalitions usually have ignorable effects and represent noise patterns. See https://anonymous.4open.science/r/ICML_rebuttal-6D88/Gogame.pdf for some new analysis.
> > >
> > > The statistical result based on a large number of cases for the ratio of coalitions that fit human intuition will be reported in the paper.
> > >
> > > Some automatically learned coalitions do not fit human intuition. As a possible explanation for this, human players typically assess patterns based on short-term tactical search and a few-step lookahead, but the value network implicitly captures long-term statistical regularities from much more games. Although these long-term patterns are difficult to interpret directly, they may reveal new shape patterns. Expert Go players say they have learned some new knowledge from these patterns.
> > >
> > > ---
> > >
> > > **Q4 “Having a bad but working baseline from your new and unstudied perspective on interactions would be greatly appreciated.”**
> > >
> > > A: A good suggestion. Unlike the mainstream of designing new attribution methods, our study of explaining the mechanism that causes the attribution conflict represent a new direction. Thus, there is no established evaluation methodology for such theoretical analysis. This is indeed a challenge.
> > >
> > > Nevertheless, we are pleased to follow your comments to conduct a new experiment for validation. Because almost all methods of constructing benchmark (ground-truth) coalitions are within the paradigm of defining a coacting group, all these benchmarks cannot be used to evaluate our theory, considering circular arguments.
> > >
> > > Instead, we compare the theoretical conflict $\phi_{conflict}$ derived from the target function $f$ based on Theorem 3.4 with the actual attribution conflict $\hat{\phi_{conflict}}$ measured in a real DNN. The DNN is learned to fit $f$. Then, $|\phi_{conflict} - \hat{\phi_{conflict}}|$ is as small as 0.0021- 0.0118, which proves the accuracy of our theory.

---

### Official Review · Reviewer_MfBL · 2025-03-14

**Overall Recommendation:** 4

**Summary:**

This paper attempts to provide insights into an issue in attributions. The issue is when one computes an attribution method, such as the Shapley value, for a coalition of inputs, it does not equal the sum of the individual input values when they are attributed to separately. This effect is explained by analyzing interactions between inputs in terms of AND-OR interactions. The paper presents results showing that the difference between the sum of individual input attributions and a group attribution can be expressed entirely in terms of a weighted sum of AND-OR interactions of non-identical coalitions that intersect with the attributed coalition. The paper provides experiments applying their explanation method on synthetic, NLP, image, and Go-related domains.

## update after rebuttal
After reviewing the rebuttal, we will maintain our score. Contrary to reviewer ZrgL, we believe the theoretical analysis the paper provides is sufficient to merit an accept. In concord with reviewer ZrgL, we believe that adding some practical applications of the theoretical results would significantly increase the impact of the paper.

**Claims And Evidence:**

Insightful theorems are provided with rigorous proofs provided in the appendix.

A variety of experimental evidence is provided.

**Essential References Not Discussed:**

Another paper on attribution of coalitions, although in the gradient based context, is given in "A Unifying Framework to the Analysis of Interaction Methods using Synergy Functions". This paper decomposes interaction methods based purely on "AND" interactions, and characterizes gradient based interactions based on their action on monomials.

**Experimental Designs Or Analyses:**

No

**Methods And Evaluation Criteria:**

In some experiments, it is not clear how faithfulness is being measures. I might suggest an insertion/deletion metric.

**Other Comments Or Suggestions:**

The above weaknesses are not necessarily killer for the paper. I recommend mentioning them and either attempting to address them or acknowledge that how to choose $\gamma_L$ still needs to be determined, but does not affect the theoretical results here.

**Other Strengths And Weaknesses:**

Strengths:
Rigorous and appropriate analysis of interactions.
This reviewer generally agrees with the direction of the paper (mathematical analysis and theorem production to gain insight of interactions among coalitions).
Generally clear writing, no typos spotted.

Weaknesses:
It appears that the decomposition of a model into AND-OR interactions is heavily sensitive to the choice of $ \gamma_L$. For example, $ \gamma_L = 0.5 v(L)$, causes all OR interactions are 0, while $ \gamma_L = -0.5 v(L)$ causes all AND interactions to be 0.

The definition of AND and OR interactions themselves are a function of the model if the LASSO method is used, which sets this method apart from other methods insofar as finding interactions now requires an optimization over the inputs for each input attributed to, as well as the calculation after $\gamma_L$ is found.

Additionally, the use of LASSO is only a suggestion, meaning it is not settled how to determine $\gamma_L$, and by extension, how to determine I_and, I_or, and derivative values.

**Questions For Authors:**

Please respond to "Strengths and Weakness"

**Relation To Broader Scientific Literature:**

The paper is firmly situated in the literature body regarding theoretical analysis of perturbation-based and game-theoretic attribution methods. The paper mentions Shapley value, Banzhaf value, Shapley-Taylor, Faith-SHAP, which are important background to these results.

**Theoretical Claims:**

No

---

> ### Author Rebuttal · Authors · 2025-04-01
>
> Thank you very much for your appreciation of this work. We would like to answer all your concerns. Please let us know if you have further questions or if you are not satisfied with the current responses. Thanks a lot.
>
> ---
>
> **Q1: In some experiments, it is not clear how faithfulness is being measured. I might suggest an insertion/deletion metric.**
>
> A: Thanks for your suggestion. In fact, there are two types of faithfulness in this research.
>
> *The first type is the faithfulness of a coalition.* In this study, we propose a new theory to analyze the faithfulness of a coalition from the perspective of attribution conflict. We propose metrics of $Q(S),R(i),R’(i)$ for evaluation. If a set of input variables mainly participate in most interactions as a group, then they will exhibit a large value of $Q(S),R(i),R’(i)$, and they can be taken as a faithful coalition; otherwise not. This evaluation strategy is motivated by the fact that the attribution conflict is a ubiquitous problem with different attribution methods, but has not been well investigated. Therefore, we propose metrics of $Q(S),R(i),R’(i)$ to evaluate the faithfulness of a coalition in terms of attribution conflict.
>
> *The second type is the faithfulness of the attribution value.* To this end, some recent theoretical studies [cite1,cite2] have mentioned that it is difficult to determine the ground-truth attribution for a DNN, so it is not a solid choice to use the network output changed by inserting/deleting a variable as the ground-truth attribution for evaluation. Specifically, these studies have pointed out that it is difficult to annotate the ground-truth attribution for the DNN. In fact, this can also be explained by our interaction theory, i.e., the inserting/deleting order significantly affects the tested attribution of a variable. For example, if the attribution mainly comes from an AND interaction S, then the deleting strategy usually assigns the interaction’s attribution to the first deleted variable in S. The inserting strategy will allocate the interaction’s attribution to the last inserted variable in S. Nevertheless, we would like to discuss more about this in the paper. Thank you very much for your constructive comments.
>
> [cite 1] Yang et al. “Benchmarking attribution methods with relative feature importance”
>
> [cite 2] Rao et al. “Towards Better Understanding Attribution Methods”
>
> ---
>
> **Q2: Another paper on attribution of coalitions, although in the gradient based context, is given in "A Unifying Framework to the Analysis of Interaction Methods using Synergy Functions".**
>
> A: Thank you very much. We are glad to cite this paper and discuss its relationship with our work. This paper introduces a unifying framework for game-theory-inspired attribution methods, analyzing feature interactions and synergy distribution to help understand which attribution method is suitable for the target model. In comparison, our study focuses on a different problem, i.e., using AND-OR interactions derived from the Möbius transform to clarify the cause for the conflict within a coalition’s attribution.
>
>
> ---
>
> **Q3: It appears that the decomposition of a model into AND-OR interactions is heavily sensitive to the choice of $\gamma_L$. … Additionally, the use of LASSO is only a suggestion, meaning it is not settled how to determine, and by extension, how to determine I_and, I_or, and derivative values.**
>
> A: This is a very good point, and the extraction of AND-OR interactions does depend on the choice of $\gamma_L$. We follow lots of previous studies[cite 1, cite 2, cite 3] to use LASSO to learn sparest AND-OR interactions, because [cite 4] has proven that if we exclusively use AND interactions for explanation, we need to use a total of $2^m$ different AND interactions to represent a single m-order OR relationship. Similarly, we need to use $2^m$ OR interactions to represent an m-order AND relationship. This theorem motivates us to use LASSO to learn sparest AND-OR interactions, and the sparest interactions are believed to capture the intrinsic representation logic of a DNN. Meanwhile, the simplicity of an explanation is another reason for us to use LASSO.
>
> Nevertheless, the optimization of $\gamma_L$ does bring some noise to the computation of interaction effects. Fortunately, we can use some optimization technologies[cite 5, cite 6] to help solve the optimization problem.
>
> [cite 1] Ren et al. “Defining and quantifying the emergence of sparse concepts in DNNs”
>
> [cite 2] Li et al. “Does a neural network really encode symbolic concepts?”
>
> [cite 3] Ren et al. “Where we have arrived in proving the emergence of sparse symbolic concepts in AI models”
>
> [cite 4] Ren et al. “Can we faithfully represent masked states to compute shapley values on a dnn?”
>
> [cite 5] Diamond et al. “CVXPY: A Python-Embedded Modeling Language for Convex Optimization”
>
> [cite 6] Agrawal et al. “A Rewriting System for Convex Optimization Problems

---

> > ### Comment · Reviewer_MfBL · 2025-04-05
> >
> > I would like to additionally note that the paper mentioned in Q2, "A Unifying Framework to the Analysis of Interaction Methods using Synergy Functions", heavily relies on the use of the mobius transform to analyze interaction methods.

---

> > > ### Author Response · Authors · 2025-04-05
> > >
> > > Thank you for your efforts in reviewing our submission and for the thoughtful feedback you provided. We appreciate you pointing this out and will be sure to emphasize the use of the Möbius transform when citing the work.

---

### Decision · Program_Chairs · 2025-05-01

**Decision:**

Accept (poster)

**Comment:**

This paper introduces a theoretical framework for analyzing coalition attributions in feature importance methods, identifying conflicts that arise when the attribution for a coalition of features does not equal the sum of the attributions of its individual components. The authors propose a formal decomposition of attribution conflicts using AND-OR interactions, offering new insights into how interactions between features result in attribution behavior. The framework generalizes to both Shapley and Banzhaf values and proposes a new metric for evaluating coalition-level faithfulness. The work is supported by theoretical analysis and experiments across synthetic functions, NLP tasks, image classification, and Go gameplay.

Strengths:
* The theoretical contribution seems strong, providing a new decomposition of attribution values using AND and OR interactions.
* The formalization reveals a common source of conflict in existing attribution methods and offers a framework for quantifying the faithfulness of feature coalitions.
* The paper unifies and extends previous formulations of Shapley and Banzhaf values,
* Reviewers MfBL and mhJ5 praised the rigor of the formalism and proofs, noting that it addresses an underexplored problem in explainable AI.
* The authors engaged deeply during the rebuttal phase, conducting new experiments (e.g., with Faith-Shap, expert validation in Go) and addressing reviewer questions about scalability, practical relevance, and potential applications.

Limitations:
* The empirical evaluation is somewhat limited in scope, especially for convincing a broader ML audience of the practical value of the theoretical results. Case studies are small-scale, largely synthetic, and not sufficient to demonstrate impact on real-world attribution workflows (though this can be argued about much of the XAI literature, too).
* Reviewer ZrgL (Reject) remains mostly unconvinced, despite the detailed rebuttal (though they did not elaborate after the authors' last comments). Their primary concerns are that:
  * The paper lacks comparisons to recent attribution methods designed for faithfulness (e.g., Faith-Shap - but this was addressed by the authors in their rebuttal),
  * The experiments are not sufficient to demonstrate utility or quantify the severity of the “attribution conflict” problem in practical settings,
* The Go experiments are anecdotal and lack evidence of benefit to expert users (but this was addressed by the authors in the rebuttal in an impressive addition).
* The definition of interactions relies on a hyperparameter optimization (via LASSO), which can bring questions about sensitivity and interpretability of the results.

Reviewer Discussions/Opinions:
* Reviewer MfBL (Accept) found the theoretical insights meaningful and believes the paper makes a valuable contribution to the understanding of attribution methods.
* Reviewer mhJ5 (Weak Accept) supports the paper for its formal strength and clear presentation of an underexplored problem, while noting that scalability and application in NLP/Vision are still open questions.
* Reviewer ZrgL (Reject) remained unconvinced after rebuttal, primarily due to limited empirical validation, lack of implementation/code, and unclear practical utility of the proposed framework. However, to my understanding, these were indeed addressed by the authors.


**Summary**
This paper offers an interesting and relevant theoretical contribution to the attribution literature, providing a novel view of coalition-level conflicts through an AND-OR interaction lens. While reviewer opinions diverged, I feel confident recommending this paper be accepted.